# The origin of carbon isotope vital effects in coccolith calcite

H.L.O. McClelland[1,2], J. Bruggeman[3], M. Hermoso[1,4] & R.E.M. Rickaby[1]

Calcite microfossils are widely used to study climate and oceanography in Earth's geological past. Coccoliths, readily preserved calcite plates produced by a group of single-celled surface-ocean dwelling algae called coccolithophores, have formed a significant fraction of marine sediments since the Late Triassic. However, unlike the shells of foraminifera, their zooplankton counterparts, coccoliths remain underused in palaeo-reconstructions. Precipitated in an intracellular chemical and isotopic microenvironment, coccolith calcite exhibits large and enigmatic departures from the isotopic composition of abiogenic calcite, known as vital effects. Here we show that the calcification to carbon fixation ratio determines whether coccolith calcite is isotopically heavier or lighter than abiogenic calcite, and that the size of the deviation is determined by the degree of carbon utilization. We discuss the theoretical potential for, and current limitations of, coccolith-based $CO_2$ paleobarometry, that may eventually facilitate use of the ubiquitous and geologically extensive sedimentary archive.

[1] Department of Earth Sciences, University of Oxford, South Parks Road, Oxford OX1 3AN, UK. [2] Department of Earth and Planetary Science, Washington University in St Louis, Campus box 1169, 1 Brookings Dr, St Louis, Missouri 63130, USA. [3] Plymouth Marine Laboratory, Prospect Place, The Hoe, Plymouth PL1 3DH, UK. [4] Équipe de Géochimie des Isotopes Stables, Institut de Physique du Globe de Paris, Sorbonne Paris Cité, Univ Paris Diderot, UMR 7154 CNRS, F-75005 Paris, France. Correspondence and requests for materials should be addressed to H.L.O.M. (email: harrym@wustl.edu).

The taxonomic delimitation of coccoliths into the isotopically light and heavy groups, where the calcite is respectively depleted and enriched in the heavy isotope of carbon compared with abiogenic calcite, is well established[1–5]. However, there have been relatively few studies of carbon isotopes in coccolith calcite[6], and fewer still where experiments have been carried out *in vivo* under closely monitored conditions[6–9]. Although so-called vital effects are known to vary with growth rate[4], and $CO_2$ availability[7], the reasons for these isotopic departures from equilibrium remain poorly understood. The relatively sparse evidence from experimental, cruise, sediment trap[10] and down-core[1–3,11] studies show that members of the light group such as *Coccolithus pelagicus* and *Calcidiscus leptoporus*, tend to be large, heavily calcifying, and have a relatively slow division rate. Contrastingly, members of the heavy group such as *Emiliania huxleyi* are generally small, lightly calcifying and with a rapid division rate[4,6,7]. The co-variance of parameters across species means that their influence upon coccolith calcite vital effects cannot be decoupled without an explicit mechanistic understanding and a quantitative model.

In coccolithophores, dissolved inorganic carbon (DIC) exists in mutually exchanging intracellular reservoirs. The intracellular process responsible for the largest isotopic fractionation in these organisms is the conversion of inorganic carbon dioxide ($CO_2$) to organic matter (carbon fixation), which has isotopic implications for carbon pools elsewhere within the cell, including, in coccolithophores, the site of calcification. The rate-limiting step of carbon fixation, catalysed by the enzyme ribulose-1,5-bisphosphate carboxylase/oxygenase (RuBisCO) proceeds far more rapidly for $CO_2$ containing the light isotope of carbon. Organic matter is therefore significantly depleted in the heavy isotope of carbon relative to the $CO_2$ source[12–14]. In addition to the kinetic isotopic discrimination effective at the enzymatic level ($\epsilon_f$), the overall carbon isotopic offset between the extracellular carbon substrate and the organic matter ($\epsilon_p$), is a function of a number of parameters that affect the isotopic composition of $CO_2$ immediately in the vicinity of the RuBisCO enzyme[15]. These parameters include growth rate ($\mu$), cell radius ($r$) and ambient $CO_2$ concentration ($[CO_2]$). So far, experiments and models attempting to understand these processes have dominantly focused on relating $\epsilon_p$ to $[CO_2]$ in phytoplankton (Table 1)[16–26], and the empirically established relationships have been extensively applied to ancient sediments to generate reconstructions of $pCO_{2atm}$ (refs 27–31). To quantitatively explain the origin of carbon isotopic vital effects in coccolith calcite, a holistic understanding of carbon isotopes within a coccolithophore cell is required, which considers the bidirectional isotopic effect of intracellular calcification and carbon fixation, and is consistent with the available biological evidence.

To understand the origin of vital effects in coccolith calcite, we derived a cellular model of carbon isotopic fluxes, whose mechanistic underpinning is based on a consideration of the available literature. The model presented here is calibrated with new data from *in vivo* experiments. We found that the so-called isotopically heavy and light group coccoliths, characterized by respectively positive and negative vital effects, are produced by coccolithophores with respectively low and high calcification to photosynthesis ratios. Vital effects are dominantly the result of competing Rayleigh-type fractionation processes and fluxes between intracellular compartments that alter the isotopic composition of carbon at the site of calcification. At the equilibrium limit, and in non-calcifying cells, our model collapses to consistency with the classic literature describing carbon isotopes in phytoplankton[16–18,21,23–25,32]. Our conclusions are compatible with observed trends in oxygen isotopes. Lastly, we discuss the potential for coccolith-based $CO_2$ paleobarometry.

## Results

**Model description.** We model the coccolithophore cell as consisting of three physical compartments: a chloroplast (where carbon fixation occurs) and a coccolith vesicle (a golgi-body derived compartment where calcification occurs) each contained within a third compartment, the cytosol (Fig. 1). Carbon within each compartment exists as $CO_2$, $HCO_3^-$ and $CO_3^{2-}$. As $HCO_3^-$ is assumed to be the substrate for calcification (justified at the rates considered here), and as membranes are assumed to be impermeable to $CO_3^{2-}$, this latter species does not feature in flux balance equations at steady state. The dynamics of the system are described entirely by $CO_2$ and $HCO_3^-$. In every compartment, $CO_2$ and $HCO_3^-$ are constantly being interconverted (Supplementary Fig. 1). Compartmental shapes and sizes relative to the cell are approximated from transition electron microscopy and assumed isomorphic across all strains and experiments. Values for the $H^+$ concentration in each compartment are taken from Anning et al.[33] The uncatalysed reacto-diffusive supply rate of $CO_2$ alone to the surface of algal cells larger than a few microns is not high enough to account for the observed rates of photosynthesis, given the low catalytic turnover and poor substrate affinity of RuBisCO[14,34–36]. The transmembrane passive diffusive supply of $CO_2$ is therefore supplemented by bicarbonate ($HCO_3^-$; the ionically charged, isotopically heavier and vastly more abundant species of DIC)[8,37,38].

Membrane permeabilities to $CO_2$ and to $HCO_3^-$ have not been measured for coccolithophores, and the isotopic fractionation of carbon by RuBisCO is known only for a single species of coccolithophore[39]. We therefore assume these parameters to be described by a set of functions whose coefficients are constant across species and treatments, and which can be constrained by empirical data. The model permits the bidirectional movement of $HCO_3^-$ and $CO_2$ across three membranes. Sinks of carbon are calcification, carbon fixation and leakage from the cell. For a full derivation of the model and complete description of the assumptions on which is it based see 'Methods' section, and Fig. 1.

In coccolithophores, bicarbonate is thought to enter the cell via a co-transport mechanism facilitated by proteins belonging to the SLC4 family of bicarbonate transporters[40,41]. These proteins, which are upregulated at low $[CO_2]$[42], couple the transmembrane movement of bicarbonate to an anion moving in the opposite direction (antiport), or a cation moving in the same direction (symport)—this is secondary active transport of bicarbonate as the cotransported ion gradient must ultimately be maintained with ATP[43]. As this type of bicarbonate transport is electroneutral, the force driving the combined transport of $HCO_3^-$ and its paired ion is dependent only on the transmembrane gradients in both ions; it is independent of the cell membrane potential. Consistent with recent experimental evidence[42], transport proteins are assumed to be upregulated when cellular carbon utilization increases, which increases the density of transport proteins in the membrane, and effectively increases the bidirectional permeability of any given membrane to $HCO_3^-$.

The model uses a flexible description of cross-membrane $HCO_3^-$ transport that can account for—but does not prescribe—facilitation by cotransporter proteins[41,42] (including countergradient transport) and upregulation of transport protein density in response to $CO_2$ limitation. Based on the results of Bach et al.[42], who showed that the transcript abundance of a putative $HCO_3^-$ transport protein increases at low DIC concentration, we here assume the effective membrane permeability to $HCO_3^-$ to be a linear function of utilization (Supplementary Fig. 2), which we define as the ratio of carbon use to passive diffusive supply of $CO_2$. Effective membrane permeability to $HCO_3^-$ includes two constants; a background value, and the utilization coefficient.

**Table 1 | Assumptions and conclusions of significant recent works modelling carbon fluxes in single-celled phytoplankton.**

| Study | Model set-up and assumptions | Conclusions |
|---|---|---|
| Holtz et al.[47,48] (E. huxleyi) | 4 compartments (PY = pyrenoid, CP = chloroplast, CV = coccolith vesicle, CY = cytosol)<br>Carbonate chemistry consists of $HCO_3^-$, $CO_3^{2-}$, $CO_2$ and $H^+$.<br>Hypothesized $Ca^{2+}/HCO_3^-$ CY-to-CV transporter coupled to an $H^+$-ATPase, with no leakage of $HCO_3^-$ from CV<br>Hypothesized upregulation of $HCO_3^-$ down-gradient flux into cell with decreased $[CO_2]$ in PY.<br>Passive $CO_2$ and $HCO_3^-$ fluxes. No $HCO_3^-$ flux from CP to PY.<br>CA assumed in CP and PY but not in CY and CV<br>Isotope model consists of 2 compartments (CY and CP) and does not consider isotopes of calcite.<br>Membrane permeabilities to $CO_2$ and $HCO_3^-$ assumed, but highly heterogeneous; different for all compartments. | $HCO_3^-$ is used at low $[CO_2]$<br>Intracellular pH gradients allow concentration of $CO_2$ around RuBisCO without up-gradient movement of carbon.<br>pHs: PY = 5.0, CY = 7.0, CP = 8.0, CV = 8.3–8.6<br>A net efflux of $CO_2$ is not necessary to remove $\delta^{13}C$ from cell |
| Bolton & Stoll,[2] (Coccolithophores) | 3 compartments (CP, CV and CY)<br>Carbonate chemistry consists of $HCO_3^-$ and $CO_2$.<br>$HCO_3^-$ fluxes are all active and independent of $[HCO_3^-]$<br>Passive $CO_2$ fluxes.<br>Membrane permeabilities and CA activities assumed from Hopkinson et al.[46].<br>$\epsilon_f$ assumed ($-27‰$). | $HCO_3^-$ active transport to CP increases at low $[CO_2]$, at the expense of $HCO_3^-$ transport to CV<br>This effect is greatest in large cells.<br>Difference in vital effects ($\delta^{13}C$ calcite—$\delta^{13}C_{medium}$) between small and large cells greatest at low $[CO_2]$ |
| Hopkinson et al.[46] (Diatoms) | 1, 2 & 3 compartments (PY, CP, CY)<br>Carbonate chemistry consists of $HCO_3^-$ and $CO_2$.<br>Used $^{18}O$ labelled DIC to track temporal evolution of carbonate system.<br>Passive $CO_2$ fluxes.<br>Passive and active $HCO_3^-$ fluxes.<br>$\epsilon_f$ assumed ($-29‰$). | Membranes are highly permeable to $CO_2$ ($1.5 \times 10^{-4}$–$5.6 \times 10^{-4}\,m\,s^{-1}$)<br>Membranes are highly impermeable to $HCO_3^-$ ($2.5 \times 10^{-8}$–$2.9 \times 10^{-7}\,m\,s^{-1}$)<br>$\delta^{13}C$ org is a function of passive diffusion of $CO_2$, active movement of $HCO_3^-$, kinetic fractionation factors associated with CA-catalysed hydration and dehydration, and the kinetic isotopic fractionation associated with RuBisCO. |
| Schulz et al.,[26] (E. huxleyi) | 2 compartments (CP and CY)<br>Carbonate chemistry consists of $HCO_3^-$ and $CO_2$.<br>Active uptake of $HCO_3^-$ and $CO_2$ independent of concentration<br>Passive $CO_2$ fluxes (membrane permeability to $CO_2 = 1.8 \times 10^{-5}$ from[76]- green algae)<br>No efflux of $HCO_3^-$<br>$\epsilon_f$ assumed ($-29‰$). | Carbon concentrating mechanism relies upon active (ATP driven) uptake of $CO_2$ and $HCO_3^-$<br>Reduction in $\epsilon_p$ with increased $HCO_3^-$ uptake into CP<br>$\epsilon_p$ is larger when there is a greater degree of intracellular carbon leakage from the chloroplast. |
| Cassar et al.,[25] (Diatom—Phaeodactylum tricornutum) | 2 compartments<br>Active and diffusive uptake of $CO_2$<br>No $HCO_3^-$ uptake<br>$\epsilon_f$ assumed ($-27‰$).<br>Inferred fluxes based on an energy minimization approach. | $\epsilon_p$ is not a unique function of $\frac{\mu}{CO_2}$<br>$\epsilon_p$ depends on leakiness of CP |
| Keller & Morel,[24] (General phytoplankton) | 1 compartment<br>No $HCO_3^-$ diffusion or efflux<br>Active $HCO_3^-$ uptake scales with growth rate<br>$\epsilon_f$ inferred from model. | Downward curvature of $\epsilon_p$ against $\frac{\mu}{CO_2}$ is consistent with active $HCO_3^-$ uptake contrary to[32]<br>$\epsilon_p$ is a poor indicator of carbon substrate |

For earlier work and the evolving appreciation of the importance of cell size, shape and growth rate see Laws et al.[15] and references therein.

Each of these constants is explicitly allowed to be zero—that is, representing zero background contribution of $HCO_3^-$ to carbon supply, and no upregulation at high utilization. As $HCO_3^-$ transport can be coupled to $Na^+$ symport or $Cl^-$ antiport via different cotransporter proteins in the SLC4 family, the transmembrane transport of $HCO_3^-$ is potentially dependent on the concentrations of these cotransported species. By describing the flux as a product of mass action (featuring the transport protein, $HCO_3^-$, and any cotransported ion species), we can capture this dependency with an asymmetric $HCO_3^-$ permeability. For simplicity, in the case of cotransporter-facilitated transport, we assume that electroneutral $Na^+$-coupled symport dominates $HCO_3^-$ transport, driven by the large extra-cellular to intra-cellular gradient of $Na^+$, and that intracellular transmembrane gradients of $Na^+$ are small. We note that the electroneutrality of this process, and thus its independence from the membrane potential, is an assumption, because some $Na^+$-coupled SLC4 transporters are known to have

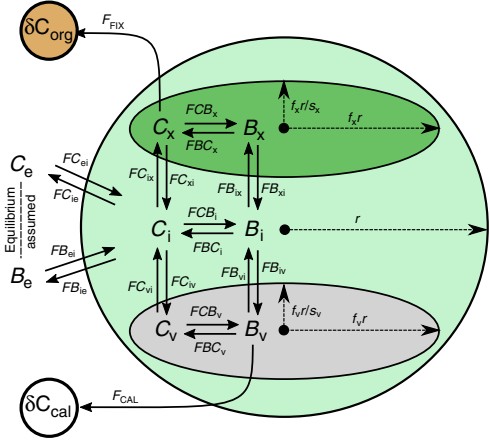

**Figure 1 | Cellular compartmental configuration and fluxes as modelled.** Chemical and isotopic equilibrium is assumed in the external medium, but neither is assumed inside the cell. Labelling of fluxes and compartments follows the convention of Hopkinson et al.[46] and Bolton and Stoll (ref. 2). Nomenclature is as follows: $C$, $B$ refer to Carbon dioxide ($CO_2$) and Bicarbonate ($HCO_3^-$) respectively. Subscripts e, i, x and v refer to compartments: external, cytosol, choroplast and coccolith vesicle respectively (NB: subscript cell refers to outer cell membrane, whilst subscript i refers to cytosol. $V_i = V_{cell} - V_x - V_v$). $F\Theta_{ab}$ represents the flux of carbon species $\Theta$, from compartment $a$ to compartment $b$ in units of mols$^{-1}$. $F\Theta\Phi_a$ represents the rate of conversion of carbon species $\Theta$, to carbon species $\Phi$, in compartment $a$, in units of mols$^{-1}$. $\Theta_a$ represents the concentration of carbon species, $\Theta$, in compartment $a$ in units of molm$^{-3}$. $P_{\Theta a}$ represents the permeability of compartment $a$ membrane to carbon species $\Theta$ in units of ms$^{-1}$. $V_a$, $SA_a$ represent the volume and surface area of compartment, $a$ in units of m$^3$ and $m^2$ respectively. $f_a$, $s_a$ represents the scale and shape factor of compartment $a$, for inferring $V_a$ and $SA_a$ ([Dimensionless]). See 'Methods' section for full model derivation and parameter values.

Na$^+$:HCO$_3^-$ stoichiometries other than 1:1, and these transporters have not yet been physiologically characterized in coccolithophores. We include a third constant representing the effective extra-cellular to intra-cellular concentration ratio of the cotransported ion species, which allows for, but does not prescribe, up-gradient bicarbonate transport across the cell membrane. Whilst being able to account for HCO$_3^-$ usage as described, the model can fall back completely to passive diffusion of $CO_2$ if warranted by observations used to constrain the universal constants.

Through isotopic discrimination at the enzymatic level, carbon fixation in the chloroplast preferentially removes the light isotope of carbon from the $CO_2$ pool, which itself is isotopically lighter than the HCO$_3^-$ pool. These effects combine to drive both chloroplast carbon pools towards isotopically heavy values. Calcification in the coccolith vesicle meanwhile preferentially removes the heavy isotope of carbon from the HCO$_3^-$ pool in the coccolith vesicle, which itself is isotopically heavier than the $CO_2$ pool. Thus calcification drives the coccolith vesicle pool towards light values. In this carbon isotope flux model, steady state is assumed whereby fluxes in and out of any given pool sum to zero, but DIC species within all compartments are allowed to be in chemical and isotopic disequilibrium. The extent of disequilibrium is a function of the rates of calcification and carbon fixation relative to those of supply and interconversion of DIC species in a given compartment. Carbonate system reactions in any given system are quantifiable functions of DIC-species-specific concentrations, H$^+$ concentration (pH), and the concentration of carbonic anhydrase (CA), the metalloenzyme

responsible for catalysing the otherwise sluggish interconversion of $CO_2$ and HCO$_3^-$ (refs 44–46) (see Supplementary Fig. 1).

**Modelling and experimental results.** The model is driven with empirical data for cell size ($r$), division rate ($\mu$), the calcification to net carbon fixation ratio (reflected in the particulate inorganic carbon to particulate organic carbon ratio; PIC:POC), external $CO_2$ concentration ([$CO_2$]) and pH, and is calibrated against the carbon isotopic deviation from DIC of the resultant organic matter ($\epsilon_O$) and calcite ($\epsilon_C$). To this end we conducted in vivo carbonate chemistry manipulation experiments, growing two strains each of two species of coccolithophore across four DIC levels. Here we present the first experimental data set that includes all of the parameters required to constrain the model.

The data and model show that coccolith calcite carbon vital effects, are positive (heavier than abiogenic calcite) when the PIC:POC ratio is low, and are negative (lighter than abiogenic calcite) when the PIC:POC ratio is high (Fig. 2). The value of PIC:POC where this divergence occurs is ~1 (Fig. 3). Vital effects are largest at high cellular carbon utilization—that is, when the cellular carbon requirement is an appreciable fraction of the carbon supply to the cell—and disappear when utilization is low. The carbon isotopic composition of organic matter meanwhile plateaus at a light value of ~ − 23 to − 24‰ (relative to DIC) at low utilizaton, and is driven towards heavier values at low utilization.

The observed phenomena are due to competing Rayleigh-type fractionation processes within the cell, which affect the isotopic composition of the bicarbonate pool in the coccolith vesicle, from which we assume calcite is precipitated. In preferentially incorporating the heavy isotope of carbon, the process of calcification drives its own substrate pool towards isotopically light values, with a greater isotopic drift manifest when the ratio of carbon usage to supply (utilization) in the coccolith vesicle is high. In the chloroplast meanwhile, the preferential removal of isotopically light carbon from the $CO_2$ pool during carbon fixation catalysed by RuBisCO, causes the isotopic composition of $CO_2$ to drift towards heavier values. The leakage of carbon from the chloroplast therefore becomes isotopically heavier, but also volumetrically decreased as the fraction of carbon supplied to the $CO_2$ pool in the chloroplast is fixed and the concentration of $CO_2$ decreases. The maximum influence of carbon fixation on the isotopic composition of the cytosolic pool, and thus on the coccolith vesicle and calcite, therefore occurs at intermediate values of $CO_2$ utilization in the chloroplast (Fig. 4). Taken to either limit; zero; and complete utilization, carbon fixation in the chloroplast has no isotopic effects on the rest of the cell.

As carbon fixation and calcification have opposing effects on the isotopic composition of the bicarbonate pool in the coccolith vesicle, the net effect on the isotopic composition of calcite is a function of the ratio of these fluxes. When the calcification to net carbon fixation ratio, recorded physically in biogenic material as the molar PIC:POC ratio, is high, the effect of calcification dominates causing vital effects to be negative. When the PIC:POC ratio is low, the influence of isotopically heavy carbon leaking from the chloroplast is greatest, resulting in positive vital effects. Vital effects are greatest when carbon utilization is high (at low [$CO_2$], or high $\mu$ or $r$), because the effective intracompartmental isotopic drift is greatest.

Cellular compartmentation and the non-infinite permeability of cellular membranes introduce the importance of intracellular heterogeneity, and the disproportionately large isotopic effect that any fractionating removal process has on its own substrate pool. Calcification, having a small fractionation factor, has very little

effect on the carbon isotopic composition of the cell as a whole, and thus organic matter, but due to vicinity, does have a large effect on the $HCO_3^-$ pool in the coccolith vesicle. By contrast, the effect of carbon fixation on the isotopic composition of calcite is significant, because although the calcification substrate pool and carbon removal by fixation are separated by two membranes, the magnitude of the fractionation factor is comparatively very large.

At the limit of chemical and isotopic equilibrium, the model behaviour becomes a function of just two variables; the utilization index ($\tau$), a compound parameter defined as the rate of cellular carbon usage to $CO_2$ supply, and PIC:POC (see 'Methods' section). At PIC:POC = 0, as in non-calcifying eukaryotic phytoplankton such as diatoms, $\tau$ is the sole variable describing the isotopic composition of organic matter. $\tau$, defined as:

$$\tau = \frac{r\rho\mu}{3P_C C_e},\qquad(1)$$

where $r$ = cell radius, $\rho$ = cellular carbon density, $\mu$ = division rate, $P_C$ = membrane permeability to $CO_2$, and $C_e$ = extracellular $CO_2$ concentration. $\tau$ is analogous to the utilization parameter described in the classic literature[19,22,32]. PIC:POC and $\tau$ are parameters that can be mathematically constrained without assuming either chemical or isotopic equilibrium, and although they do not describe the system in its entirety when not at equilibrium, they are nevertheless useful for describing the system in a meaningful two-dimensional space (Fig. 3).

## Discussion

Carbon isotopic compositions of coccolith calcite, and of organic matter, can be fully explained via competing Rayleigh-type fractionation processes within the compartmentalised cell (Fig. 4). The recently observed vanishing coccolith vital effect at high $[CO_2]$[7,9] naturally emerges as a result of the model, as Rayleigh-type fractionation is at a minimum when the rate of supply of carbon far exceeds usage.

There is an evolving consensus that $HCO_3^-$ forms a larger proportion of the carbon entering the cell at low $[CO_2]$[26,47,48], a conclusion supported by a transcriptional upregulation of a putative $HCO_3^-$ cotransporter protein at low $[DIC]$[42], and directly measured $CO_2$ and $HCO_3^-$ uptake rates in *E. huxleyi*[8,37,38]. Experimental and molecular evidence suggests that in coccolithophores, the transmembrane supply of $HCO_3^-$ is facilitated by transporters belonging to the solute carrier 4 (SLC4) family of cotransporter proteins[40,49–51], which facilitate $Cl^-/HCO_3^-$ antiport, or $Na^+/HCO_3^-$ symport. Inside the cell, we assume that the transmembrane concentration gradients of $Cl^-$ and $Na^+$ ions are negligible. In this scenario, trans-membrane $HCO_3^-$ fluxes are proportional to the $HCO_3^-$ concentration on the proximal side of the membrane; the concentrations of cotransported ion species are incorporated in the effective permeability to $HCO_3^-$. However, at the cell membrane, we allow the extra-cellular to intra-cellular concentration ratio of the cotransported ion species to vary, thus allowing up-gradient movement of $HCO_3^-$. Due to the concentrations of both $Cl^-$ and $Na^+$ likely being higher outside

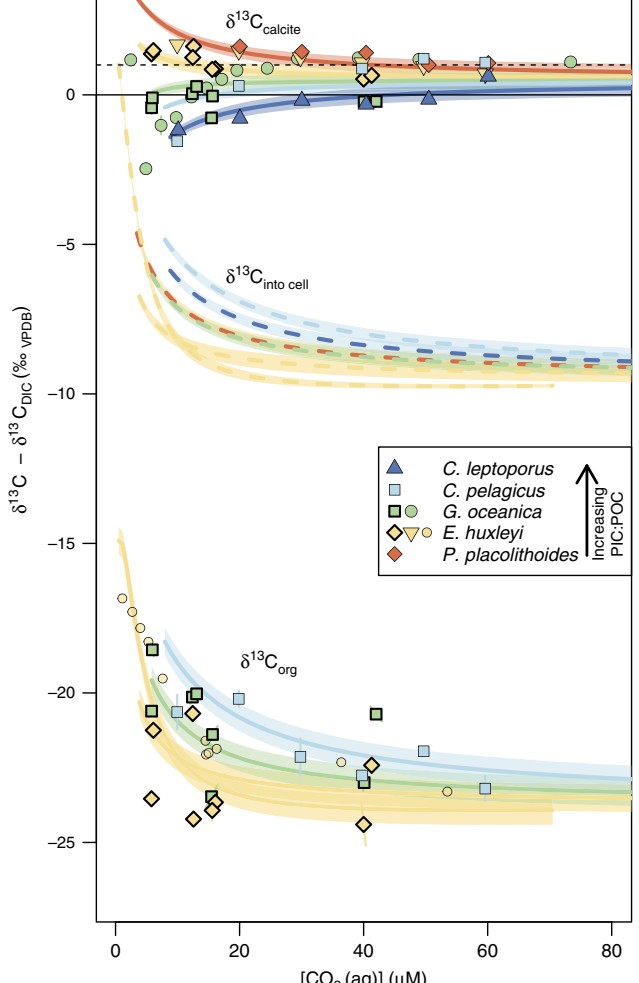

**Figure 2 | Model output.** Model output across the experimental range of $CO_2$, with empirical data superimposed. The black horizontal dashed line shows the expected composition of abiogenic calcite. Universal constants used to generate model output are constrained by the points outlined in bold, here re-sampled 1,000 times from distributions defined by the associated analytical uncertainties. The data from this study also include additional *G. oceanica* data (calcite only; green filled circles) and organic carbon isotopic compositions for *C. pelagicus* from samples from a previous study[9]. See Supplementary Data 1 for values. Additional literature data are from dilute batch experiments manipulated by varying DIC[9] (calcite carbon isotopic compositions only: *C. leptoporus*—dark blue upwards triangles; *C. pelagicus*—blue squares; *E. huxleyi*—yellow downwards triangles; and *P. placolithoides*—orange diamonds), and by varying pH (ref. 75; *E. huxleyi*—organic carbon isotopic compositions only: small yellow filled circles). Further data are superimposed in Supplementary Fig. 3. Data points are all the average of two replicates, with the ends of bars representing the value of each replicate. Solid lines and shaded envelopes represent respectively the mean and standard deviation of isotopic compositions predicted by the model for the 1,000 repeated calibrations. Values for compartmental pH are taken from published measurements[33]. Different species are represented by a representative set of parameters, which for ease of illustration are held constant with varying $[CO_2]$. Representative values are taken from across a range of sources in the literature and from our own unpublished data (cell radius, division rate, PIC:POC are as follows: *E. huxleyi* – 2.3 µm, 1 day$^{-1}$, 0.5; *G. oceanica* - 3.5 µm, 1 day$^{-1}$, 1.1; *C. pelagicus* – 8 µm, 0.9 day$^{-1}$, 1.2; *C. leptoporus* – 6 µm, 0.65 day$^{-1}$, 2.2; and *P. placolithoides* – 7 µm, 0.8 day$^{-1}$, 0.3). Strong co-variance between cell size or growth rate with $CO_2$ would not be well represented in this projection. Note: for the pH manipulated model output $HCO_3^-$ uptake is enhanced at low $CO_2$, manifest as a more rapid isotopic enrichment at low $CO_2$ (steep yellow model curve). All species curves are a similar shape controlled by a number of trade-offs, and with the point of inflexion and maximum $\delta^{13}C_{calcite}$ determined by varying combinations of cell size, PIC:POC and growth rate. There are some regions of parameter space that are theoretically possible, but may not be observed in reality due to poor growing conditions. Regions of model space not populated by data should be treated as hypothetical.

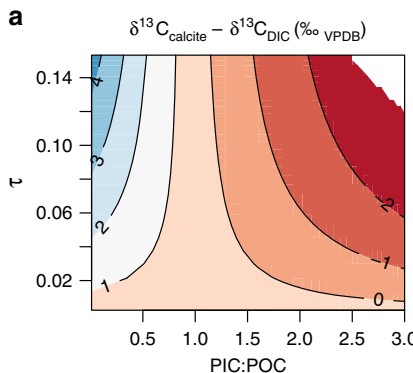
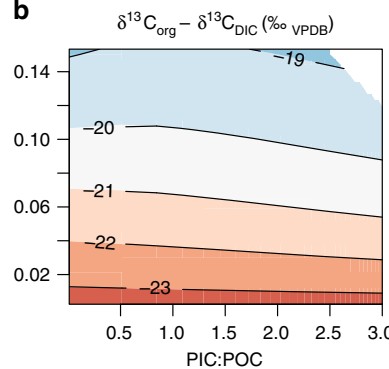

**Figure 3 | Effect of cellular utilization and PIC:POC on isotopes.** Summary of model output—isotopic compositions as a function of PIC:POC and of the utilization parameter, $\tau$. $\tau$ approximates utilization, and is a dimensionless linear function of growth rate, cell size [DIC], as defined in equation (1). (**a**) Calcite carbon isotopic compositions depend strongly on PIC:POC—if PIC:POC is high, calcite becomes isotopically light as utilization increases, if PIC:POC is low, calcite becomes isotopically heavy as utilization increases. (**b**) The carbon isotopic composition of organic matter is largely independent of PIC:POC, and is a strong function of the utilization parameter.

the cell than in the cytosol, we assume that symporting proteins dominate $HCO_3^-$ transport.

Other mechanisms for $HCO_3^-$ transport have been proposed which have implications for the isotopic composition of coccolith calcite. It has been suggested that negative carbon isotopic vital effects in coccolith calcite are due to an increased pumping of $HCO_3^-$ into the chloroplast at low $[CO_2]$, at the expense of moving $HCO_3^-$ to the coccolith vesicle with no diffusive efflux of $HCO_3^-$ from any compartment, leaving the cytosolic pool, and thus the coccolith vesicle, isotopically light in carbon[2]. We are aware of no molecular evidence to date of putative primary active transport proteins associated with bicarbonate transport in this group of organisms however. Furthermore, prohibiting $HCO_3^-$ efflux from any cellular compartment has significant implications for the isotopic composition of intracellular carbon pools, and imposes an artificially high importance on the ratio of inward fluxes of $HCO_3^-$ to $CO_2$. Although effective membrane permeabilities to $CO_2$ or $HCO_3^-$ have not been measured directly in coccolithophores, the values constrained by our model (permeability to $CO_2 = \sim 9.3 \times 10^{-4}\,ms^{-1}$, and to $HCO_3^-$ being 3–4 orders of magnitude lower) are consistent with those calculated for the closely related group, the diatoms[46] (see 'Methods' section).

In some unicellular algae, CA is emitted from the cell to accelerate sluggish kinetics of $HCO_3^-$ dehydration in the boundary layer. Despite reports that *E. huxleyi* lacks external CA $(CA_e)$[37], more recent work argues for the presence[52] and possibly upregulation of $CA_e$ at low $[CO_2]$[41,42]. $CA_e$ was not however detected in either *Gephyrocapsa oceanica* nor *C. pelagicus*, even when cells were carbon limited[49], though more research on species other than *E. huxleyi* is required. Here we assume that $CA_e$ is present in high enough concentrations that the external surface of the cell is in chemical and isotopic equilibrium with the bulk seawater. If future work concludes that $CA_e$ is absent in one or more species, it may be necessary to allow chemical and isotopic disequilibrium in the cellular boundary layer (for implications of this scenario see Supplementary Fig. 4 and Supplementary Note 1).

The model described here does not separately represent the pyrenoid: a compartment of low pH nested within the relatively alkaline chloroplast where $CO_2$ and RuBisCO are thought to be highly concentrated. The value of the effective modelled isotopic discrimination factor for RuBisCO, describing the difference between the isotopic compositon of $CO_2$ in the chloroplast and that of organic matter $(\epsilon_{fe})$, is therefore effectively a pyrenoid/

RuBisCO black-box fractionation for the given set-up. The extent to which $\epsilon_{fe}$ is representative of the enzymatic RuBisCO fractionation, $\epsilon_f$, is a critical question regarding the validity of using *in vitro* measured values of $\epsilon_f$ in models that do not have a detailed representation of the pyrenoid. The introduction of an additional $CO_2$—permeable membrane between the ambient medium and the site of carbon fixation, may allow a further isotopic discrimination to take place by the preferential uptake of $CO_2$ over $HCO_3^-$, which would allow the value of $\epsilon_f$ to be even smaller than the value of $\epsilon_{fe}$ predicted by our model—thus providing an explanation when there is a discrepancy between low values of $\epsilon_f$ and high values of $\epsilon_p$. However, conversely, if compartmental utilization of the pyrenoid were very high, $\epsilon_f$ may be far higher than the value of $\epsilon_{fe}$ predicted by the model. Although isotopic fractionation by RuBisCO has been extensively studied in plants, only two studies to date have estimated the *in vitro* isotopic fractionation of carbon associated with carbon fixation catalysed by the type of RuBisCO (form ID) found in the 'red-lineage algae' such as the coccolithophores and the diatoms[39,53], and these values are surprisingly low (11 and 19‰ respectively). Despite this, recent modelling work has continued to use the markedly larger fractionation factors associated with plants and cyanobacteria (Table 1). The fitted value for $\epsilon_{fe}$ output by our model is $\sim 14$–15‰.

The calcite saturation state in the coccolith vesicle $(\Omega_{calcite})$ is controlled by $[CO_3^{2-}]$ and $[Ca^{2+}]$, and by the solubility product of calcite in seawater $(K_{sp})$, which is a function of salinity, temperature and pressure[44,54]. At most $CO_2$ concentrations, the model suggests that the coccolith vesicle is oversaturated with respect to calcite (Supplementary Fig. 5). However, at $CO_2$ concentrations below around $30\,\mu M$, the model predicts that $\Omega_{calcite}$ falls below unity, which would usually render calcite thermodynamically unstable. A number of possibilities may individually, or in combination, lead to a precipitation-promoting $\Omega_{calcite}$ inside the CV at this lower $CO_2$ range. First, measuring pH in an intracellular compartment is difficult, and is fraught with uncertainties, so the pH in the coccolith vesicle may be significantly higher than the previously measured value of 7.1 (ref. 33), increasing the fraction of DIC present in the form $CO_3^{2-}$. Second, $HCO_3^-$ transport to the coccolith vesicle may be enhanced at low $CO_2$ – this could be an upregulation of transport proteins[42] specific to the coccolith vesicle membrane, or, as suggested elsewhere, an energy-driven net $Ca^{2+}/HCO_3^-$ co-transport mechanism[48]. Third, the solubility product $(K_{sp})$ of calcite is two orders of magnitude higher (that is, $\Omega_{calcite}$ is lower

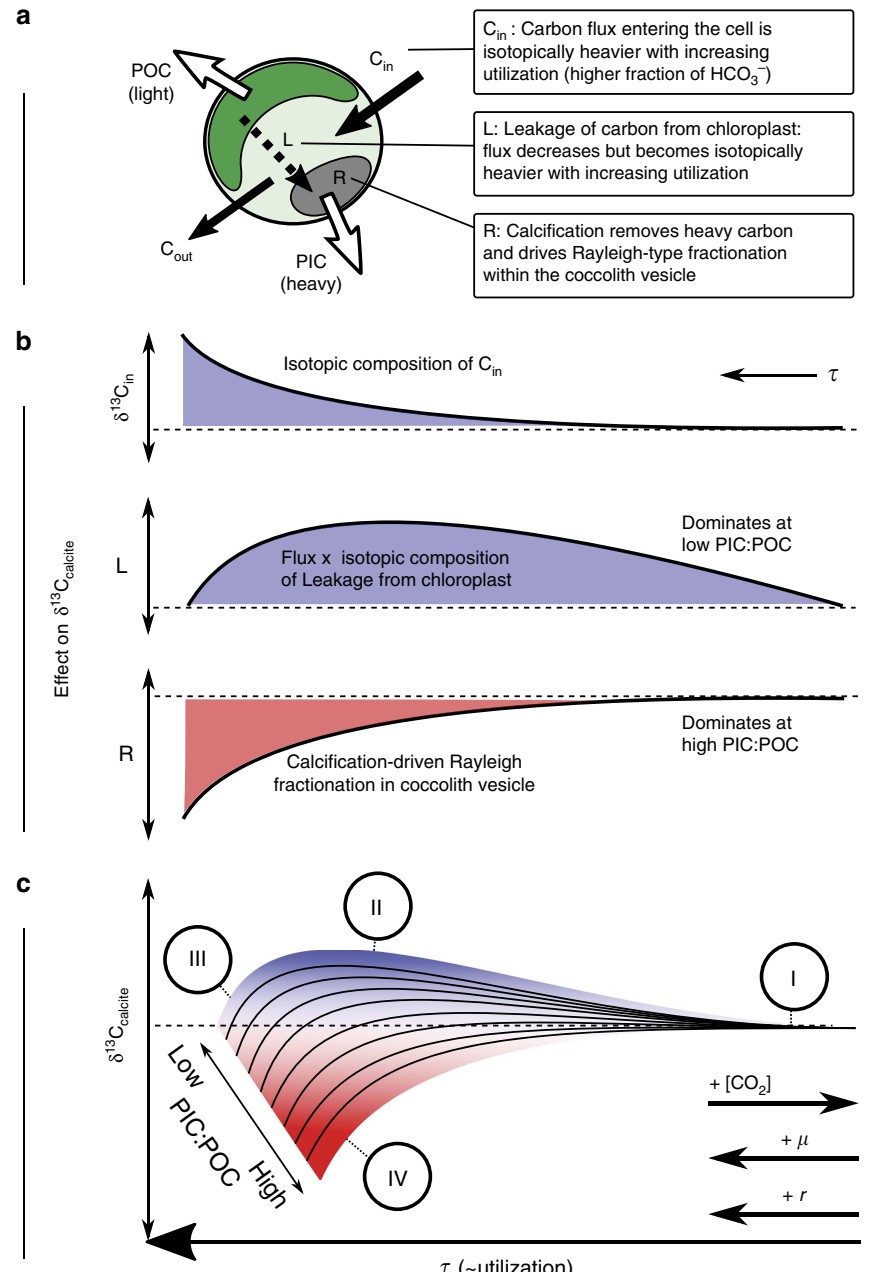

**Figure 4 | Summary of causes of vital effects.** Summary of causes of carbon isotope vital effects in coccolith calcite. (**a**) Coccolithophore cell with isotopically depleted POC produced in the chloroplast and isotopically enriched PIC produced in the coccolith vesicle. (**b**) Schematic representation of the effect of three dominant processes affecting the isotopic composition of calcite ($\delta^{13}C_{calcite}$). (**c**) Net result of carbon isotopic vital effects in coccolith calcite. At low $\tau$ (that is, high $CO_2$, low cell size or low growth rate), the flux of carbon into the cell far exceeds intracellular processes, and all species plateau at the same value (I). The value of this plateau is heavier than that of the carbon entering the cell due to intracellular interconversion of carbon species and preferential loss of light carbon dioxide. Low PIC:POC: for a cell of low PIC:POC, as $\tau$ increases (I–II), the flux of isotopically heavy carbon leaving the chloroplast (depleted in light carbon by carbon fixation catalysed by RuBisCO) influences the isotopic composition of the coccolith vesicle. As $\tau$ increases further (II–III), the flux of carbon from the chloroplast, although isotopically increasingly heavy, tends towards zero, and Rayleigh-type fractionation within the coccolith vesicle itself (which removes isotopically heavy carbon) takes over, driving the pool to light values. High PIC:POC: For a cell of high PIC:POC, as $\tau$ increases (I–IV) the Rayleigh-type fractionation due to calcite precipitation drives the isotopic composition of the coccolith vesicle to light values. The effect of heavy carbon leaking from the chloroplast is obscured. Across all cells of all PIC:POC values, the trends in vital effects are moderated by the biological up-regulation of $HCO_3^-$ transport proteins, which increases overall permeability to carbon and increases the and isotopic composition of carbon entering the cell with increasing $\tau$, and which dampens the Rayleigh fractionation-type effects when $\tau$ is high.

and calcite precipitation is less likely to occur) in water with a salinity typical of seawater compared with distilled water[54]. Although the salinity of the CV is likely to be close to that of seawater due to the osmotic pressure that is associated with large salinity differences[55], maintaining a lower salinity in the coccolith vesicle is a possible mechanism of elevating $\Omega_{calcite}$. Fourth, through forming temporary bonds to $Ca^{2+}$ ions in solution, acidic polysaccharides, which are known to be present in the CV,

and throughout the coccolith calcite lattice, strongly influence calcification, and may cause the localized $\Omega_{calcite}$ to be significantly higher than that of the bulk CV. Values of $[Ca^{2+}]$ in the CV have not been measured directly with much success[33,56], but hypothetical estimates vary across an order of magnitude (0.5–4 mM)[47,57,58]. The localized calcite precipitation-inducing effect on carbonate chemistry may even allow coccoliths to be built in a mostly calcite under-saturated vesicle. Little work has been undertaken to understand the localized influences of polysaccharides on carbonate chemistry however.

Under the modelled range of $[CO_2]$, $CO_2$ of the chloroplast is at a level comparable with the Michaelis–Menten half-saturation constant of RuBisCO typical for a coccolithophore ($\sim 30$–40 µM $CO_2$; Supplementary Fig. 5)[34,39], as might be expected for maximum efficiency. At low $[CO_2]$, $CO_2$ in the chloroplast does drop significantly below this value, however, the pyrenoid/thylakoid complex is a small region of very low pH where RuBisCO and $CO_2$ are in far higher concentration than in the rest of the chloroplast[34,59]. Locally therefore, the low $[CO_2]$ inferred here for the chloroplast is unlikely to be reflective of the environment immediately in the vicinity of the RuBisCO enzyme.

Oxygen kinetic isotopic fractionation factors (KIFs) for the hydration and dehydration of $CO_2$ and $HCO_3^-$ respectively are not known, and theoretical predictions vary widely[60,61]. As a result, this model cannot currently be fully resolved with respect to oxygen isotopes. An additional complexity over carbon isotopes arises with oxygen because oxygen in DIC exchanges with oxygen in water, which even at high DIC concentrations is many orders of magnitude more abundant[44,62,63]. The rate of oxygen exchange with water and the residence time of DIC species in a compartment is therefore critical to the preservation of vital effects in calcite. As coccolith calcite exhibits vital effects in oxygen, the cell cannot be in complete isotopic equilibrium with respect to oxygen. Across coccolithophores, a positive correlation is seen between the carbon and oxygen isotopic compositions of calcite (Supplementary Fig. 6a)[9]. However, although $HCO_3^-$ is isotopically heavier than $CO_2$ in carbon, it is lighter than $CO_2$ in oxygen, so the observed trends in carbon and oxygen isotopes cannot be simultaneously explained solely by a shift in cellular carbon source from dominantly $HCO_3^-$ to $CO_2$ or vice versa. We hypothesize that the increased difference in oxygen isotopic vital effects between large and small cells observed at at low $[CO_2]$[2–5,9] is due to a difference in the oxygen isotopic composition of DIC entering the cell following the increased influx of bicarbonate at high utilization (Supplementary Fig. 6b). We further hypothesize that this signal may be superimposed on a Rayleigh-type fractionation, which drives the combined $HCO_3^-/CO_3^{2-}$ pool in the coccolith vesicle to heavy values at high utilization, as the isotopically lighter (in oxygen) $CO_3^{2-}$ is taken up into calcite, and by CA-catalysed hydration and dehydration which pushes the system further towards equilibrium at high DIC (Supplementary Fig. 6c).

The model of vital effects in coccolith calcite presented here not only accounts for previously unexplained variations in isotopic compositions, but also provides the theoretical framework for a new suite of proxies. The ultimate ambitious aim is to accurately quantify $pCO_{2atm}$ in the geological past, but if the whole system can be constrained, this would additionally yield information aspects of contemporaneous coccolithophore physiology. Coccolithophores include the only family of organisms known to be responsible for producing alkenones—the most widely used organic molecule extracted from ancient sediments for estimating past $pCO_{2atm}$ via isotopic analysis. In these analyses, traditionally isotopic compositions have been related to $[CO_2]$, and thus $pCO_{2atm}$, through univariate and empirical

relationships[27–31]. Here we have shown that the system is more complicated than this, and the isotopic fractionation of carbon into calcite (and to a lesser degree organic material) is strongly bivariate in PIC:POC and $\tau$ space even when chemical and isotopic equilibrium can be assumed (Fig. 3). With paired measurements of $\epsilon_C$ and $\epsilon_O$ from analysis of ancient sediments, in theory our model can be used to iteratively search parameter space to minimize the misfit between observed and predicted isotopic compositions, and thus simultaneously predict the most likely values of PIC:POC and $\tau$ of these ancient organisms. We recently extracted acidic polysaccharides from within the calcite lattice of large ancient coccoliths[64], which opens up the possibility of isotopically characterizing non-alkenone producing species, and may, in the future, supersede alkenones as the target molecule for organic carbon isotopic analyses. If, in addition, an independent estimate of PIC:POC were obtained[65], the difference between the carbon isotopic compositions of organic matter and calcite alone could be used to constrain the system, circumventing the need for an independent estimate of the isotopic composition of DIC (Fig. 5). To extract an estimate of $[CO_2]$ from the constrained value of $\tau$, growth rate and cell size must be known. The difference in isotopic composition of calcite produced by different sizes of coccolithophore cells growing in the same seawater may also be used to constrain $[CO_2]$, but independent estimates of PIC:POC, growth rate and cell size for both coccolithophore cell size fractions would be necessary (Fig. 5). For some species of coccolithophore, cell size can be estimated from coccolith size[11,65], and PIC:POC can be estimated from the cross-sectional aspect ratio of a coccolith[65]. Strontium/calcium ratios in coccolith calcite have also been proposed to reflect growth rate[66,67]. At present, however, there are large statistical uncertainties associated with all of these estimates, which are multiplicative when propagated. Our novel mechanistic insight provides a biologically grounded theoretical basis, but improvements in estimates of these other integral parameters are necessary before accurate quantitative coccolith-based paleo-$CO_2$ barometry will be possible.

## Methods

**Culture experiments.** Culture experiments were undertaken in the Department of Earth Sciences at the University of Oxford. Duplicate monoclonal batch cultures of four strains of coccolithophore belonging to the family Noëlaerhabdaceae were grown in sterile filtered (0.2 µm) artificial seawater prepared according to ESAW[68] adapted for a range of DIC concentrations ((DIC) = 1.380 mM, 2.147 mM, 3.067 mM and 6.135 mM) at constant pH total scale (8.2) by varying sodium bicarbonate addition and titration with HCl and with nitrate (442 µM), phosphate (5.00 µM), vitamins, trace metals and ethylenediaminetetraacetic acid according to K/2 (ref. 69). Cultures were maintained at 15 °C with an incident photon flux of 250 µE and a 12/12 light/dark cycle. Cells were acclimated for >20 generations in dilute batch culture for each experimental condition before inoculation. Cells were inoculated in 2.4 l polycarbonate flasks, with no headspace and sealed off to the air with teflon lined caps. Removal of medium during the experiment was unavoidable due to the need to count and measure cells, and resulted in a maximum headspace of 20 cm$^3$ at harvest. To minimize the drift in culture conditions throughout the course of the experiment, cells were harvested at $\sim 1$–2% (and never >4%) of maximum cell density, which was determined for each experimental condition and strain combination via prelimenary experimentation. Strains were AC478 (RCC1211 *G. oceanica* from Portuguese coast in Atlantic Ocean), AC472 (RCC1216 *E. huxleyi*, from Tasman Sea in Pacific Ocean), AC448 (RCC1256 *E. huxleyi*, Icelandic coast in Atlantic Ocean) and AC279 (RCC1314 *G. oceanica*, French coast in Atlantic Ocean) from the Roscoff culture collection (RCC). Particulate material was harvested by dry filtration onto pre-weighed membranes with 0.2 µm pore-size, and rinsed of salt with a minimal amount of neutralized deionised water. Coccolithophore size and concentration were obtained using a Beckman Z2 Coulter Counter. Coccosphere and cell size were measured three times each respectively pre- and post-decalcification both morning and evening on the harvest day and the preceding day. Cells were decalcified by reducing the pH of the suspension with HCl addition to 5.0 with for around 20 min. The Coulter counter was calibrated to use ESAW + K/2 medium as an electrolyte, and for use with the acidified electrolyte, to accommodate for the difference in ionic strength. Cell division is synchronized under the light/dark cycle and cell size was assumed to increase linearly throughout the day[70]. By measuring cell and coccosphere size

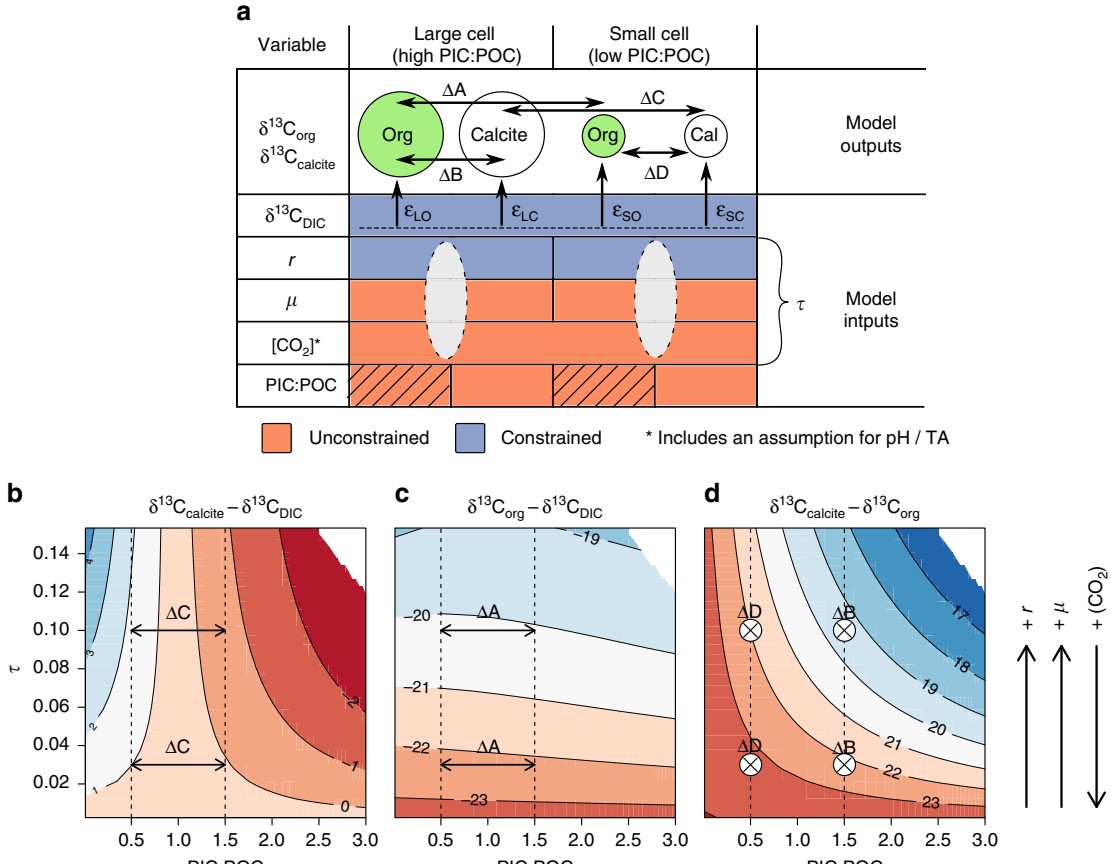

**Figure 5 | Concept for interpretation of sedimentary data. (a)** Once calibrated, the model provides values of isotopic compositions of organic matter and of calcite ($\delta^{13}C_{org}$ and $\delta^{13}C_{calcite}$ respectively) as outputs, given a number of input parameters ($\delta^{13}C_{DIC}$, $r$, $\mu$, [$CO_2$] and PIC:POC). $\delta^{13}C_{DIC}$ can be estimated from foraminifera. $r$, $\mu$ and [$CO_2$] form a single compound variable for each size fraction (semi-transparent ovals), $\tau$ (equation (1)), which cannot be deconvolved without additional evidence. The model can be inverted and used to iteratively search parameter space for the values of $\tau$ and PIC:POC that minimize the misfit between model-predicted and measured isotopic compositions. $r$ can be estimated from coccolith size. Boxes without a division denote a shared variable between measured isotopic compositions. $\epsilon_O$ is only weakly dependent on PIC:POC so to a first approximation, $\tau$ may be estimated from $\epsilon_O$ alone (as shown by shaded box). The assumption included in [$CO_2$] concerns the ambient concentration of $HCO_3^-$, but the model output is highly insensitive to this assumption. **(b–d)** Model output describing theoretical carbon isotopic compositions of inorganic (**b**) and organic (**c**) material relative to $\delta^{13}C_{DIC}$, and $\delta^{13}C_{calcite}$—$\delta^{13}C_{org}$ (**d**) over a range of PIC:POC and $\tau$. The double headed arrows point to the difference between a cell with a PIC:POC typical of *E. huxleyi* (left) and a typical *C. pelagicus* (right); the number of $\delta^{13}C_{calcite}$ contours crossed decreases with decreasing $\tau$.

morning and evening, the bias introduced due to the time of day of measurement can be removed by interpolation to the same time of day. This also removes the daily variation of cell size[70]. Culture health was monitored by cell counts and microscope inspection on alternate days. Molar PIC and POC were measured with a Rock Eval analyser in the Earth Sciences department at Oxford University.

**Isotope measurements.** Carbon and oxygen isotopic compositions of calcite were measured at the University of Oxford using a VG Isogas Prism II mass spectrometer with an on-line VG Isocarb common acid bath preparation system. Samples were first rinsed with neutralized deionised water to remove any salt. Samples were then dosed with acetone and dried at 60 °C for at least 30 min. In the instrument they were reacted with purified phosphoric acid at 90 °C. Calibration to PDB standard was via the international standard NBS-19 using the Oxford in-house (NOCZ) Carrara marble standard. Reproducibility of replicated standards was better than 0.1‰ ($1\sigma$) for $\delta^{13}C$ and $\delta^{18}O$ expressed relative to the V-PDB standard. Carbon isotopic composition of organic material was measured on an automated carbon and nitrogen elemental analyzer (Carlo Erba EA1108) at the Research Laboratory for Archaeology and the History of Art at the University of Oxford. Samples were decalcified with HCl, and rinsed with MilliQ water at least three times before being weighed into tin capsules. The internal alanine standard reproducibility is ~0.16‰ expressed relative to the V-PDB standard.

**Assumptions of the model.** The assumptions on which this model is based, are:

- Cells are in steady-state growth. The concentration of every carbon pool (that is, each carbon species in each intracellular compartment) is constant, which implies that all fluxes associated with that pool sum to zero.
- The cell and compartments are assumed to be isomorphic across all strains and conditions.
- $CO_2$ and $HCO_3^-$ ions move across membranes via passive diffusion and facilitated diffusion respectively. Both chemical species are assumed to move across membranes with a flux proportional to surface area of the membrane and the concentration of the carbon species on the proximal side of the membrane. There is no concentration-independent active uptake of bicarbonate or carbon dioxide. Membranes are impermeable to carbonate ions. There is no other source of carbon to the cell.
- Loss of carbon from the cell is via carbon fixation, calcification, passive diffusion of $CO_2$, and co-transport of $HCO_3^-$.
- The membrane permeability to $CO_2$ is constant.
- The effective membrane permeability to $HCO_3^-$, which incorporates the concentrations of coported ions, is a linear function of background carbon utilization ($U_0$; defined in equation (2)), representing the ability of cells to upregulate transport proteins. The special cases whereby $HCO_3^-$ permeability is non-adaptive, or is zero are explicitly allowed.
- The constants describing the permeability of membranes to $CO_2$ and to $HCO_3^-$ are identical across all species, and for all membranes except the cell membrane where the strong gradient of $Na^+$ from the extra-cellular to intra-cellular environment is assumed to dominate $HCO_3^-$ transport via $Na^+$ coport, thus introducing an asymmetric permeability. As this is an electroneutral process, the membrane potential does does affect this flux, which is assumed to be a product of mass-action.

- pH in each compartment is constant, and is prescribed *a priori*, according to the values measured by Anning *et al.*[33].
- Organic matter is not remobilised from the organic pool to the cytosol via mitochondrial respiration.
- Movement of carbon across membranes does not impart a kinetic isotopic fractionation.
- All cells are assumed to exhibit active external CA (see Supplementary Note 1).
- The chloroplast is assumed to consist of a single compartment with no pyrenoid. The large isotopic fractionation of carbon that occurs during fixation catalysed by the enzyme RuBisCO ($\epsilon_f$), is here factored into the model as an effective pyrenoid/RuBisCO black box fractionation ($\epsilon_{fe}$).

**Membrane permeabilities.** A putative bicarbonate transport protein has been observed to be upregulated of at low $[CO_2]$[42]. Here these results are interpreted as consistent with an approximately linear increase in transcript abundance with the degree of carbon utilization (Supplementary Fig. 2). This physiological result is assumed to reflect an increase in synthesis of $HCO_3^-$ transport proteins, and thus the density of transport proteins in the membrane, manifest as an increase in the effective permeability of membranes to $HCO_3^-$. The flux of $HCO_3^-$ through membranes facilitated by SCL4-like exchange proteins is proportional the the product of $HCO_3^-$ on the proximal side of the membrane and $Na^+$ on the proximal side of the membrane for coport, and that of $Cl^-$ on the distal side of the membrane for antiport. For simplicity, we assume $HCO_3^-$ transport to be driven by $Na^+$ coport, and the gradients of the coported ion to be negligible across all membranes, except the cellular membrane where it is substantial. This assumption is factored into the model via an additional universal constant, which describes the extra-cellular to intra-cellular concentration gradient of the coported ion. This assumption is more coherent with the current biological literature than equivalent assumptions of previous models, including concentration-independent implicitly ATP-driven active uptake of $HCO_3^-$, assumed by Keller *et al.*[24] to scale with growth rate, or by Bolton and Stoll[2] to be independent of all other parameters.

In this model, the permeability of membranes to $CO_2$ ($P_{Ccell}$) is assumed to be constant. The permeability of membranes to $HCO_3^-$ ($P_{Bcell}$) is assumed to increase linearly with utilization of background (that is, before upregulation of anion exchange proteins) carbon supply ($U_0$), which is defined as:

$$U_0 = \frac{F_{FIX} + F_{CAL}}{C_{in0}}. \quad (2)$$

where $C_{in0}$ is the net carbon supply when membrane permeabilities to $CO_2$ and to $HCO_3^-$ are at their background values, and $F_{FIX} + F_{CAL}$ is the rate of fixation of carbon into organic and inorganic matter. The membrane permeability to bicarbonate is therefore:

$$P_{Bcell} = T_0 + T_U U_0 \quad (3)$$

$$= T_0 + T_U \frac{F_{FIX} + F_{CAL}}{(C_e P_{Ccell} + GB_e T_0) SA_{cell}}, \quad (4)$$

where $T_0$ is the base-level membrane permeability to $HCO_3^-$ and $T_U$ is the gradient of the increase in membrane permeability to $HCO_3^-$ with increasing utilization; $F_{FIX}$ and $F_{CAL}$ are the rates of carbon fixation and calcification respectively, $C_e$ and $B_e$ are the concentrations of $CO_2$ and $HCO_3^-$ in the ambient medium respectively, and $SA_{cell}$ is the surface area of the cell. The large $Na^+$ gradient at the cell membrane is included as an additional constant, $G$. This line of reasoning is a set of assumptions factored into the model via four universal constants describing the membrane permeability to $CO_2$ and to $HCO_3^-$ ($P_{Ccell}$, $T_0$, $T_U$ and $G$). These four constants will be constrained by the data, explicitly leaving open the option of non-upregulated $HCO_3^-$ transport (through $T_U = 0$), complete impermeability of membranes to $HCO_3^-$ (through $T_0 = T_U = 0$), and through up- and down-gradient movement of $HCO_3^-$ dependent on the inferred $Na^+$ ion gradient.

**Compartment shapes and sizes.** Each intracellular compartment is assumed to be an oblate spheroid. The equatorial axis of the spheroid, $a$, is assumed to have a constant ratio with the cell radius. This is referred to as the scaling factor ($s f_c \times r = a_c$). The polar axis, $c$, is assumed to have a constant ratio with the equatorial axis. This ratio is referred to as the aspect ratio factor ($a f_c \times a_c = c_c$). Isometry is assumed across species, and in this way, two constants are used to describe the relative size and shape of each compartment, and therefore their volumes and surface areas. The surface area of an oblate spheroid is given by:

$$S_{oblate} = 2\pi a^2 \left(1 + \frac{1-e^2}{e} \tanh^{-1} e\right) \quad \text{where} \quad e^2 = 1 - \frac{c^2}{a^2}. \quad (5)$$

The volume of a spheroid is given by:

$$V = \frac{4\pi a^2 c}{3} \quad (6)$$

**Intracellular carbonic anhydrase.** The CAs are a family of zinc-containing metalloenzymes responsible for catalysing the hydration and dehydration of $CO_2$ and $HCO_3^-$ respectively. Their behaviour is well described by Michaelis–Menten

kinetics[45]. Given the general form of an enzymatically catalysed reaction:

$$E + S \underset{k_r}{\overset{k_f}{\rightleftharpoons}} ES \overset{k_{cat}}{\rightarrow} E + P \quad (7)$$

where $E$, $S$, $ES$ and $P$ denote enzyme, substrate, enzyme–substrate complex and product concentrations, respectively. $k_f$, $k_r$ and $k_{cat}$ are rate constants respectively for the binding and unbinding of substrate to the enzyme, and for the maximum catalytic throughput of the reaction.

According to the Michaelis–Menten equations, the velocity of a reaction such as equation (7) is given by:

$$v = \frac{d[P]}{dt} = V_{max} \frac{[S]}{K_m + [S]} = k_{cat}[E] \frac{[S]}{K_m + [S]}, \quad (8)$$

where $K_m$ is the Michaelis constant, or half-saturation constant ($K_m = \frac{k_r + k_{cat}}{k_f}$), and describes the substrate concentration when $v = \frac{V_{max}}{2}$. When $[S] \ll K_m$, equation (8) becomes:

$$v \approx \frac{k_{cat}}{K_m}[E] \cdot [S], \quad (9)$$

and $k_{cat}/K_m$ is the specific activity of the enzyme in units of $M^{-1} s^{-1}$. This is analogous to a rate constant ($k_p$), or activity, when multiplied by the concentration of the enzyme. The rate constant for CA-catalysed hydration of $CO_2$ is then:

$$k_{pCA} = \frac{k_{cat}}{K_m}\Big|_{CA} [CA], \quad (10)$$

As enzymes catalyse reactions but do not alter the position of equilibrium, the rate constant for the reverse reaction, CA-catalysed dehydration of $HCO_3^-$, is given by the hydration rate constant divided by the equilibrium constant:

$$k_{mCA} = \frac{k_{pCA}}{K_1^*}. \quad (11)$$

A value of $2.7 \times 10^7 M^{-1} s^{-1}$ was determined by Uchikawa *et al.*[45] for the $\frac{k_{cat}}{K_m CA}$ of the hydration reaction catalysed by bovine erythrocyte CA.

**Model derivation.** Fluxes of $CO_2$ through membranes are the product of membrane surface area (SA), $CO_2$ concentration on the source side of the membrane, and membrane permeability to the specific carbon species. From Fig. 1 expressions for transmembrane $CO_2$ fluxes are:

$$
\begin{aligned}
FC_{ei} &= C_e P_{Ccell} SA_{cell} \\
FC_{ie} &= C_i P_{Ccell} SA_{cell} \\
FC_{ix} &= C_i P_{Cx} SA_x \\
FC_{xi} &= C_x P_{Cx} SA_x \\
FC_{iv} &= C_i P_{Cv} SA_v \\
FC_{vi} &= C_v P_{Cv} SA_v,
\end{aligned}
\quad (12)
$$

and likewise for $HCO_3^-$ fluxes:

$$
\begin{aligned}
FB_{ei} &= B_e G P_{Bcell} SA_{cell} \\
FB_{ie} &= B_i P_{Bcell} SA_{cell} \\
FB_{ix} &= B_i P_{Bx} SA_x \\
FB_{xi} &= B_x P_{Bx} SA_x \\
FB_{iv} &= B_i P_{Bv} SA_v \\
FB_{vi} &= B_v P_{Bv} SA_v,
\end{aligned}
\quad (13)
$$

where all fluxes are defined in Fig. 1. $G$ is the constant that describes the effective asymmetry of the cell membrane to $HCO_3^-$ due to the transmembrane $Na^+$ gradient.

In each compartment, carbon dioxide and bicarbonate ions are interconverted by the reversible hydration and hydroxylation reactions, and by hydration in the presence of CA:

$$
\begin{aligned}
CO_2 + H_2O &\underset{k_{m1}}{\overset{k_{p1}}{\rightleftharpoons}} H^+ + HCO_3^- \quad &\text{(hydration)} \\
CO_2 + OH^- &\underset{k_{m4}}{\overset{k_{p4}}{\rightleftharpoons}} HCO_3^- \quad &\text{(hydroxylation)} \\
CO_2 + H_2O &\underset{k_{mCA}}{\overset{k_{pCA}}{\rightleftharpoons}} H^+ + HCO_3^- \quad &\text{(CA catalysed hydration)}.
\end{aligned}
\quad (14)
$$

where arrow annotations denote rate constants (see section for CA rate constants). Combining the rate constants of equation (14) at constant pH and [CA] in a

generic compartment, $a$, gives:

$$k_{CBa} = OH_a \cdot k_{p4} + k_{p1} + k_{pCA} \cdot CA_a \qquad (15)$$

$$k_{BCa} = k_{m4} + H_a \cdot (k_{m1} + k_{mCA} \cdot CA_a) \qquad (16)$$

where $k_{CBa}$ denotes the reaction rate constant for the conversion of $CO_2$ to $HCO_3^-$, and $k_{BCa}$ denotes the rate constant for the reverse reaction. $H_a$, $OH_a$ and $CA_a$ refer to the concentrations of $H^+$ ions, $OH^-$ ions and CA in compartment, $a$. As compartment-specific pH and [CA] are defined *apriori*, $k_{CBa}$ and $k_{BCa}$ can be treated as pH and [CA] dependent compound rate constants. The rates of the reactions described by equation (14) in each compartment are therefore:

$$\begin{aligned}
FCB_i &= V_i C_i k_{CBi} \\
FBC_i &= V_i B_i k_{BCi} \\
FCB_x &= V_x C_x k_{CBx} \\
FBC_x &= V_x B_x k_{BCx} \\
FCB_v &= V_v C_v k_{CBv} \\
FBC_v &= V_v B_v k_{BCv}
\end{aligned} \qquad (17)$$

Carbon fluxes throughout the cell can be fully described with equations (12), (13) and (17), and with two additional output fluxes; the rate of calcification ($F_{CAL}$) and the rate of photosynthetic carbon fixation ($F_{FIX}$; Fig. 1). Assuming steady state, the rate of change of the amount of each carbon species in each compartment is zero, giving:

$$\begin{aligned}
V_i \frac{dC_i}{dt} &= 0 = FC_{ei} + FC_{vi} + FC_{xi} + FBC_i - FC_{ie} - FC_{iv} - FC_{ix} - FCB_i \\
V_i \frac{dB_i}{dt} &= 0 = FB_{ei} + FB_{vi} + FB_{xi} + FCB_i - FB_{ie} - FB_{iv} - FB_{ix} - FBC_i \\
V_x \frac{dC_x}{dt} &= 0 = FC_{ix} + FBC_x - FCB_x - FC_{xi} - F_{FIX} \\
V_x \frac{dB_x}{dt} &= 0 = FB_{ix} + FCB_x - FBC_x - FB_{xi} \\
V_v \frac{dC_v}{dt} &= 0 = FC_{iv} + FBC_v - FCB_v - FC_{vi} \\
V_v \frac{dB_v}{dt} &= 0 = FB_{iv} + FCB_v - FBC_v - FB_{vi} - F_{CAL}
\end{aligned} \qquad (18)$$

At steady state these differential equations become linear functions of the concentration of the different inorganic carbon species. Substituting in equations (12), (13) and (17):

$$\begin{aligned}
-C_e P_{Ccell} SA_{cell} &= C_v P_{Cv} SA_v + C_x P_{Cx} SA_x + B_i V_i k_{BCi} \\
&\quad + C_i(-P_{Ccell} SA_{cell} - P_{Cv} SA_v - P_{Cx} SA_x - V_i k_{CBi}) \\
-B_e GP_{Bcell} SA_{cell} &= B_v P_{Bv} SA_v + B_x P_{Bx} SA_x + C_i V_i k_{CBi} \\
&\quad + B_i(-P_{Bcell} SA_{cell} - P_{Bv} SA_v - P_{Bx} SA_x - V_i k_{BCi}) \\
F_{FIX} &= C_i P_{Cx} SA_x + B_x V_x k_{BCx} + C_x(-V_x k_{CBx} - P_{Cx} SA_x) \\
0 &= B_i P_{Bx} SA_x + C_x V_x k_{CBx} + B_x(-V_x k_{BCx} - P_{Bx} SA_x) \\
0 &= C_i P_{Cv} SA_v + B_v V_v k_{BCv} + C_v(-V_v k_{CBv} - P_{Cv} SA_v) \\
F_{CAL} &= B_i P_{Bv} SA_v + C_v V_v k_{CBv} + B_v(-V_v k_{BCv} - P_{Bv} SA_v).
\end{aligned} \qquad (19)$$

This set of interdependent equations can be written as a linear system of the form:

$$\mathbf{A} \cdot \mathbf{\Phi} = \mathbf{N}, \qquad (20)$$

where $\mathbf{A}$ is the coefficient matrix of the linear system, $\mathbf{N}$ is the nonhomogeneous term and $\mathbf{\Phi}$ is the unknown vector containing the carbon species concentrations in each compartment. Equation (20) can be solved for $\mathbf{\Phi}$, where $\mathbf{A}$ and $\mathbf{N}$, defined by the dynamic carbon species concentration equations, are as follows:

$$\mathbf{A} = \begin{pmatrix}
\begin{matrix} -P_{Ccell}SA_{cell} \\ -P_{Cv}SA_v \\ -P_{Cx}SA_x \\ -V_i k_{CBi} \end{matrix} & V_i k_{BCi} & P_{Cx}SA_x & 0 & P_{Cv}SA_v & 0 \\
V_i k_{CBi} & \begin{matrix} -P_{Bcell}SA_{cell} \\ -P_{Bv}SA_v \\ -P_{Bx}SA_x \\ -V_i k_{BCi} \end{matrix} & 0 & P_{Bx}SA_x & 0 & P_{Bv}SA_v \\
P_{Cx}SA_x & 0 & \begin{matrix} -V_x k_{CBx} \\ -P_{Cx}SA_x \end{matrix} & V_x k_{BCx} & 0 & 0 \\
0 & P_{Bx}SA_x & V_x k_{CBx} & \begin{matrix} -V_x k_{BCx} \\ -P_{Bx}SA_x \end{matrix} & 0 & 0 \\
P_{Cv}SA_v & 0 & 0 & 0 & \begin{matrix} -V_v k_{CBv} \\ -P_{Cv}SA_v \end{matrix} & V_v k_{BCv} \\
0 & P_{Bv}SA_v & 0 & 0 & V_v k_{CBv} & \begin{matrix} -V_v k_{BCv} \\ -P_{Bv}SA_v \end{matrix}
\end{pmatrix}$$

$$\mathbf{N} = \begin{pmatrix}
-C_e P_{Ccell} SA_{cell} \\
-B_e GP_{Bcell} SA_{cell} \\
F_{FIX} \\
0 \\
0 \\
F_{CAL}
\end{pmatrix}$$

$$\mathbf{\Phi} = \begin{pmatrix}
C_i \\
B_i \\
C_x \\
B_x \\
C_v \\
B_v
\end{pmatrix}$$

[$^{13}$C] is very low compared to [$^{12}$C], carbon fluxes are assumed to represent $^{12}$C. $^{13}$C dynamics are therefore determined by prefixing each $C$ flux with corresponding $R$, where:

$$R = \frac{[^{13}C]}{[^{12}C]}.$$

By definition,

$$R = R_{standard}\left(\frac{\delta^{13}C}{1,000} + 1\right),$$

so the assumption of balanced growth allows both offset $R_{standard}$ and scale factor $R_{standard}\frac{R_{standard}}{1,000}$ to be eliminated while preserving the same system of equations. Effectively, we can thus prefix each flux directly with the associated $\delta^{13}C$ value. For fractionation fluxes, this leads to a prefix with the sum of the $\delta^{13}C$ of the source pool and the process-specific fractionation factor ($\epsilon$). In the following, $\delta_{\Theta a}$ refers to the $\delta^{13}C$ of carbon species, $\Theta$, in compartment, $a$.

The following fractionation factors (in ‰)[44], are assumed constant:

$$\begin{aligned}
\epsilon_{p1} &= -13 \\
\epsilon_{p4} &= -11 \\
\epsilon_{m1} &= -22 \\
\epsilon_{m4} &= -20 \\
\epsilon_{pCA} &= -1 \\
\epsilon_{mCA} &= -10,
\end{aligned} \qquad (21)$$

where the subscripts are consistent with those from equation (14). Analogous to the compound rate constants of equations (15) and (16), the interconversion of carbon dioxide and bicarbonate has a pH and [CA] dependent, compartment-specific, compound isotopic fractionation factor.

$$\epsilon_{CBa} = \frac{\epsilon_{cb4} \cdot OH_a \cdot k_{p4} + \epsilon_{cb1} \cdot k_{p1} + \epsilon_{cbCA} \cdot k_{pCA} \cdot CA_a}{OH_a \cdot k_{p4} + k_{p1} + k_{pCA} \cdot CA_a} \qquad (22)$$

$$\epsilon_{BCa} = \frac{\epsilon_{bc4} \cdot k_{m4} + \epsilon_{bc1} \cdot H_a \cdot k_{m1} + \epsilon_{bcCA} \cdot k_{pCA} \cdot CA_a}{k_{m4} + \epsilon_{bc1} \cdot H_a \cdot (k_{m1} + k_{mCA} \cdot CA_a)}. \qquad (23)$$

When the linear system described for carbon fluxes is solved, all carbon species concentrations in all compartments are known. Dynamic equations for the isotopic composition of each compartment, assuming balanced growth, and expressed in terms of carbon dioxide, and bicarbonate fluxes and fractionation factors are thus:

$$\begin{aligned}
V_i \frac{d\delta_{Ci} \cdot C_i}{dt} = 0 &= \delta_{Ce} \cdot FC_{ei} + \delta_{Cv} \cdot FC_{vi} + \delta_{Cx} \cdot FC_{xi} + (\delta_{Bi} + \epsilon_{BCi}) \cdot FBC_i \\
&\quad + \delta_{Ci} \cdot (-FC_{ie} - FC_{iv} - FC_{ix}) - (\delta_{Ci} + \epsilon_{CBi}) \cdot FCB_i
\end{aligned} \qquad (24)$$

$$\begin{aligned}
V_i \frac{d\delta_{Bi} \cdot B_i}{dt} = 0 &= \delta_{Be} \cdot FB_{ei} + \delta_{Bv} \cdot FB_{vi} + \delta_{Bx} \cdot FB_{xi} + (\delta_{Ci} + \epsilon_{CBi}) \cdot FCB_i \\
&\quad + \delta_{Bi} \cdot (-FB_{ie} - FB_{iv} - FB_{ix}) - (\delta_{Bi} + \epsilon_{BCi}) \cdot FBC_i
\end{aligned} \qquad (25)$$

$$\begin{aligned}
V_x \frac{d\delta_{Cx} \cdot C_x}{dt} = 0 &= \delta_{Ci} \cdot FC_{ix} + (\delta_{Bx} + \epsilon_{BCx}) \cdot FBC_x - (\delta_{Cx} + \epsilon_{CBx}) \cdot FCB_x \\
&\quad - \delta_{Cx} \cdot FC_{xi} - (\delta_{Cx} + \epsilon_{FIX}) \cdot F_{FIX}
\end{aligned} \qquad (26)$$

$$\begin{aligned}
V_x \frac{d\delta_{Bx} \cdot B_x}{dt} = 0 &= \delta_{Bi} \cdot FB_{ix} + (\delta_{Cx} + \epsilon_{CBx}) \cdot FCB_x - (\delta_{Bx} + \epsilon_{BCx}) \cdot FBC_x \\
&\quad - \delta_{Bx} \cdot FB_{xi}
\end{aligned} \qquad (27)$$

$$\begin{aligned}
V_v \frac{d\delta_{Cv} \cdot C_v}{dt} = 0 &= \delta_{Ci} \cdot FC_{iv} + (\delta_{Bv} + \epsilon_{BCv}) \cdot FBC_v - (\delta_{Cv} + \epsilon_{CBv}) \cdot FCB_v \\
&\quad - \delta_{Cv} \cdot FC_{vi}
\end{aligned} \qquad (28)$$

$$\begin{aligned}
V_v \frac{d\delta_{Bv} \cdot B_v}{dt} = 0 &= \delta_{Bi} \cdot FB_{iv} + (\delta_{Cv} + \epsilon_{CBv}) \cdot FCB_v - (\delta_{Bv} + \epsilon_{BCv}) \cdot FBC_v \\
&\quad - \delta_{Bv} \cdot FB_{vi} - (\delta_{Bv} + \epsilon_{CAL}) \cdot F_{CAL}
\end{aligned} \qquad (29)$$

At steady state the differential equations become linear functions of $\delta^{13}C$ of the different carbon species in the different compartments. Substituting in equations

(12), (13) and (17):

$$\begin{aligned}
\epsilon_{CBi} \cdot V_i C_i k_{CBi} - \delta_{Ce} \cdot C_e P_{Ccell} SA_{cell} - \epsilon_{BCi} \cdot V_i B_i k_{BCi} \\
= \delta_{Cv} \cdot C_v P_{Cv} SA_v + \delta_{Cx} \cdot C_x P_{Cx} SA_x + \delta_{Bi} \cdot V_i B_i k_{BCi} \\
+ \delta_{Ci} \cdot (-C_i P_{Ccell} SA_{cell} - C_i P_{Cv} SA_v - C_i P_{Cx} SA_x - V_i C_i k_{CBi})
\end{aligned} \quad (30)$$

$$\begin{aligned}
\epsilon_{BCi} \cdot V_i B_i k_{BCi} - \delta_{Be} \cdot B_e GP_{Bcell} SA_{cell} - \epsilon_{CBi} \cdot V_i C_i k_{CBi} \\
= \delta_{Bv} \cdot B_v P_{Bv} SA_v + \delta_{Bx} \cdot B_x P_{Bx} SA_x + \delta_{Ci} \cdot V_i C_i k_{CBi} \\
+ \delta_{Bi} \cdot (-B_i P_{Bcell} SA_{cell} - B_i P_{Bv} SA_v - B_i P_{Bx} SA_x - V_i B_i k_{BCi})
\end{aligned} \quad (31)$$

$$\begin{aligned}
\epsilon_{CBx} \cdot V_x C_x k_{CBx} - \epsilon_{BCx} \cdot V_x B_x k_{BCx} + \epsilon_{FIX} \cdot F_{FIX} \\
= \delta_{Ci} \cdot C_i P_{Cx} SA_x + \delta_{Bx} \cdot V_x B_x k_{BCx} \\
+ \delta_{Cx} \cdot (-V_x C_x k_{CBx} - C_x P_{Cx} SA_x - F_{FIX})
\end{aligned} \quad (32)$$

$$\begin{aligned}
\epsilon_{BCx} \cdot V_x B_x k_{BCx} - \epsilon_{CBx} \cdot V_x C_x k_{CBx} \\
= \delta_{Bi} \cdot B_i P_{Bx} SA_x + \delta_{Cx} \cdot V_x C_x k_{CBx} \\
+ \delta_{Bx} \cdot (-V_x B_x k_{BCx} - B_x P_{Bx} SA_x)
\end{aligned} \quad (33)$$

$$\begin{aligned}
\epsilon_{CBv} \cdot V_v C_v k_{CBv} - \epsilon_{BCv} \cdot V_v B_v k_{BCv} \\
= \delta_{Ci} \cdot C_i P_{Cv} SA_v + \delta_{Bv} \cdot V_v B_v k_{BCv} \\
+ \delta_{Cv} (-V_v C_v k_{CBv} - C_v P_{Cv} SA_v)
\end{aligned} \quad (34)$$

$$\begin{aligned}
\epsilon_{BCv} \cdot V_v B_v k_{BCv} - \epsilon_{CBv} \cdot V_v C_v k_{CBv} + \epsilon_{CAL} \cdot F_{CAL} \\
= \delta_{Bi} \cdot B_i P_{Bv} SA_v + \delta_{Cv} \cdot V_v C_v k_{CBv} \\
+ \delta_{Bv} \cdot (-V_v B_v k_{BCv} - B_v P_{Bv} SA_v - F_{CAL})
\end{aligned} \quad (35)$$

This set of interdependent equations can also be written as a linear system of the form:

$$\mathbf{A} \cdot \mathbf{\Phi} = \mathbf{N}, \quad (36)$$

where $\mathbf{A}$ is the coefficient matrix of the linear system, $\mathbf{N}$ is the nonhomogeneous term and $\mathbf{\Phi}$ is the unknown vector containing the carbon isotopic compositions of each carbon species in each compartment. Equation (36) can be solved for $\mathbf{\Phi}$, where $\mathbf{A}$ and $\mathbf{N}$, defined by the dynamic carbon species concentration equations, are as follows:

$$\mathbf{A} = \begin{pmatrix}
\begin{matrix} -C_i P_{Ccell} SA_{cell} \\ -C_i P_{Cv} SA_v \\ -C_i P_{Cx} SA_x \\ -V_i C_i k_{CBi} \end{matrix} & V_i B_i k_{BCi} & C_x P_{Cx} SA_x & 0 & C_v P_{Cv} SA_v & 0 \\
V_i C_i k_{CBi} & \begin{matrix} -B_i P_{Bcell} SA_{cell} \\ -B_i P_{Bv} SA_v \\ -B_i P_{Bx} SA_x \\ -V_i B_i k_{BCi} \end{matrix} & 0 & B_x P_{Bx} SA_x & 0 & B_v P_{Bv} SA_v \\
C_i P_{Cx} SA_x & 0 & \begin{matrix} -V_x C_x k_{CBx} \\ -C_x P_{Cx} SA_x \\ -F_{FIX} \end{matrix} & V_x B_x k_{BCx} & 0 & 0 \\
0 & B_i P_{Bx} SA_x & V_x C_x k_{CBx} & \begin{matrix} -V_x B_x k_{BCx} \\ -B_x P_{Bx} SA_x \end{matrix} & 0 & 0 \\
C_i P_{Cv} SA_v & 0 & 0 & 0 & \begin{matrix} -V_v C_v k_{CBv} \\ -C_v P_{Cv} SA_v \end{matrix} & V_v B_v k_{BCv} \\
0 & B_i P_{Bv} SA_v & 0 & 0 & V_v C_v k_{CBv} & \begin{matrix} -B_v P_{Bv} SA_v \\ -V_v B_v k_{BCv} \\ -F_{CAL} \end{matrix}
\end{pmatrix}$$

$$\mathbf{N} = \begin{pmatrix}
\epsilon_{CBi} \cdot V_i C_i k_{CBi} - \delta_{Ce} \cdot C_e P_{Ccell} SA_{cell} - \epsilon_{BCi} \cdot V_i B_i k_{BCi} \\
\epsilon_{BCi} \cdot V_i B_i k_{BCi} - \delta_{Be} \cdot B_e GP_{Bcell} SA_{cell} - \epsilon_{CBi} \cdot V_i C_i k_{CBi} \\
\epsilon_{CBx} \cdot V_x C_x k_{CBx} - \epsilon_{BCx} \cdot V_x B_x k_{BCx} + \epsilon_{FIX} \cdot F_{FIX} \\
\epsilon_{BCx} \cdot V_x B_x k_{BCx} - \epsilon_{CBx} \cdot V_x C_x k_{CBx} \\
\epsilon_{CBv} \cdot V_v C_v k_{CBv} - \epsilon_{BCv} \cdot V_v B_v k_{BCv} \\
\epsilon_{BCv} \cdot V_v B_v k_{BCv} - \epsilon_{CBv} \cdot V_v C_v k_{CBv} + \epsilon_{CAL} \cdot F_{CAL}
\end{pmatrix}$$

$$\mathbf{\Phi} = \begin{pmatrix} \delta_{Ci} \\ \delta_{Bi} \\ \delta_{Cx} \\ \delta_{Bx} \\ \delta_{Cv} \\ \delta_{Bv} \end{pmatrix}$$

**Equilibrium limit special case.** At the limit where CA activity is infinitely high in all compartments, complete chemical and isotopic equilibrium with respect to carbon can be assumed. In this scenario, the total [DIC] (D) in each compartment and its isotopic composition, along with (prescribed) pH, completely specifies the concentrations and isotopic compositions of the different inorganic carbon species in each compartment.

The dynamic equations for total DIC (D) in the three different compartments are:

$$V_i \frac{dD_i}{dt} = 0 = FC_{ei} - FC_{ie} + FB_{ei} - FB_{ie} - FC_{ix} + FC_{xi} \\ - FB_{ix} + FB_{xi} - FC_{iv} + FC_{vi} - FB_{iv} + FB_{vi} \quad (37)$$

$$V_x \frac{dD_x}{dt} = 0 = FC_{ix} - FC_{xi} + FB_{ix} - FB_{xi} - F_{FIX} \quad (38)$$

$$V_v \frac{dD_v}{dt} = 0 = FC_{iv} - FC_{vi} + FB_{iv} - FB_{vi} - F_{CAL} \quad (39)$$

In steady state all three equations must equal 0, which implies:

$$F_{FIX} = FC_{ix} - FC_{xi} + FB_{ix} - FB_{xi} \quad (40)$$

$$F_{CAL} = FC_{iv} - FC_{vi} + FB_{iv} - FB_{vi} \quad (41)$$

Inserting these into equation (37), the expression for cytoplasm total DIC, gives:

$$V_i \frac{dD_i}{dt} = FC_{ei} - FC_{ie} + FB_{ei} - FB_{ie} - F_{FIX} - F_{CAL} = 0. \quad (42)$$

The four fluxes across the cell membrane equal:

$$FC_{ei} = P_C C_e SA \quad (43)$$

$$FC_{ie} = P_C C_i SA \quad (44)$$

$$FB_{ei} = P_B GB_e SA \quad (45)$$

$$FB_{ie} = P_B B_i SA \quad (46)$$

where the permeability for $HCO_3^-$ relates to the permeability for $CO_2$ through background utilization:

$$P_B = P_C (T_0 + T_U U_0). \quad (47)$$

As a result, all cross-membrane fluxes are proportional to $P_C SA$ and equation (42) can be divided through by this factor. With chemical equilibrium and prescribed $pH_c$ in the cytoplasm, $B_c$ is proportional to $C_c$ for any compartment $c \in \{e, i, x, v\}$:

$$B_c = \frac{K^*_{H_2CO_3}}{H_c} C_c \quad (48)$$

where $K^*_{H_2CO_3}$ is the temperature and salinity-dependent chemical equilibrium constant, and $H_c$ the $H^+$ concentration ($H_c = 10^{-pH_c}$). Thus equation (42) can be rewritten as:

$$V_i \frac{dD_i}{dt} = C_e \left(1 + G \frac{K^*_{H_2CO_3}}{H_e}(T_0 + T_U U_0)\right) - C_i \left(1 + \frac{K^*_{H_2CO_3}}{H_i}(T_0 + T_U U_0)\right) \\ - \frac{F_{FIX}}{P_C SA} - \frac{F_{CAL}}{P_C SA} = 0. \quad (49)$$

Isolating $C_i$ and divide through by $C_e$ gives:

$$\frac{C_i}{C_e} = \frac{\left(1 + G\frac{K^*_{H_2CO_3}}{H_e}(T_0 + T_U U_0)\right) - \frac{F_{FIX}}{C_e P_C SA} - \frac{F_{CAL}}{C_e P_C SA}}{1 + \frac{K^*_{H_2CO_3}}{H_i}(T_0 + T_U U_0)} \quad (50)$$

Thus, concentrations and fluxes of carbon species in all compartments are known.

Following the same process, two additional fluxes across the chloroplast membrane equal:

$$FC_{ix} = f_x P_C C_i SA \quad (51)$$

$$FC_{xi} = f_x P_C C_x SA \quad (52)$$

where $f_x$ is the ratio of the product of cellular membrane permeability and surface area to that of the chloroplast. Equation (40) can therefore be rewritten:

$$\frac{F_{FIX}}{f_x P_C SA C_e} = \frac{C_i}{C_e}\left(1 + \frac{K^*_{H_2CO_3}}{H_i}(T_0 + T_U U_0)\right) - \frac{C_x}{C_e}\left(1 + \frac{K^*_{H_2CO_3}}{H_x}(T_0 + T_U U_0)\right), \quad (53)$$

and inserting equation (50) becomes:

$$\frac{C_x}{C_e} = \frac{\left[\frac{\left(1 + G\frac{K^*_{H_2CO_3}}{H_e}(T_0 + T_U U_0)\right) - \frac{F_{FIX}}{C_e P_C SA} - \frac{F_{CAL}}{C_e P_C SA}}{1 + \frac{K^*_{H_2CO_3}}{H_i}(T_0 + T_U U_0)}\right]\left(1 + \frac{K^*_{H_2CO_3}}{H_i}(T_0 + T_U U_0)\right) - \frac{F_{FIX}}{f_x P_C SA C_e}}{1 + \frac{K^*_{H_2CO_3}}{H_x}(T_0 + T_U U_0)}. \quad (54)$$

Similarly, two additional fluxes across the coccolith vesicle membrane equal:

$$FC_{iv} = f_v P_C C_i SA \quad (55)$$

$$FC_{vi} = f_v P_C C_v SA \quad (56)$$

where $f_v$ is the ratio of the product of cellular membrane permeability and surface

area to that of the coccolith vesicle. Equation (41) can therefore be rewritten:

$$\frac{F_{CAL}}{f_v P_C SAC_e} = \frac{C_i}{C_e}\left(1 + \frac{K^*_{H_2CO_3}}{H_i}(T_0 + T_U U_0)\right) - \frac{C_v}{C_e}\left(1 + \frac{K^*_{H_2CO_3}}{H_v}(T_0 + T_U U_0)\right),$$

(57)

and inserting equation (50) likewise becomes:

$$\frac{C_v}{C_e} = \frac{\left[\frac{\left(1 + G\frac{K^*_{H_2CO_3}}{H_e}(T_0 + T_U U_0)\right) - \frac{F_{FIX}}{C_e P_C SA} - \frac{F_{CAL}}{C_e P_C SA}}{1 + \frac{K^*_{H_2CO_3}}{H_i}(T_0 + T_U U_0)}\right]\left(1 + \frac{K^*_{H_2CO_3}}{H_i}(T_0 + T_U U_0)\right) - \frac{F_{FIX}}{f_v P_C SAC_e}}{1 + \frac{K^*_{H_2CO_3}}{H_v}(T_0 + T_U U_0)}.$$

(58)

The values $f_x$ and $f_v$ are constants when all membranes are assumed to have the same permeability, and isometry is assumed.

Membrane permeability to bicarbonate is reliant on the background utilization of carbon, $U_0$. The value of $U_0$ is given by:

$$U_0 = \frac{F_{FIX} + F_{CAL}}{P_C SA(C_e + T_0 GB_e)} = \frac{\frac{F_{FIX}}{P_C SA} + \frac{F_{CAL}}{P_C SA}}{C_e + T_0 GB_e} = \frac{\frac{F_{FIX}}{P_C SAC_e} + \frac{F_{CAL}}{P_C SAC_e}}{1 + T_0 G\frac{K^*_{H_2CO_3}}{H_e}}$$

(59)

At equilibrium, $HCO_3^-$ has a constant isotopic offset from $CO_2$, denoted here as $E$. As both $HCO_3^-$ and $CO_2$ in each compartment are known if one is known, the whole system can be described by $CO_2$ alone. The dynamic equations for isotopic fluxes are:

$$\frac{d\delta C_i}{dt} = 0 = \delta C_e FC_{ei} - \delta C_i FC_{ie} + \delta C_x FC_{xi} - \delta C_i FC_{ix} + \delta C_v FC_{vi}$$
$$- \delta C_i FC_{iv} - (\delta C_i + E)FB_{ie} + (\delta C_e + E)FB_{ei} + (\delta C_x + E)FB_{xi} - (\delta C_i + E)FB_{ix}$$
$$+ (\delta C_v + E)FB_{vi} - (\delta C_i + E)FB_{iv}$$

(60)

$$\frac{d\delta C_x}{dt} = 0 = \delta C_i FC_{ix} - \delta C_x FC_{xi} + (\delta C_i + E)FB_{ix} - (\delta C_x + E)FB_{xi} - (\delta C_x + \epsilon_f)F_{FIX}$$

(61)

$$\frac{d\delta C_v}{dt} = 0 = \delta C_i FC_{iv} - \delta C_v FC_{vi} + (\delta C_i + E)FB_{iv} - (\delta C_v + E)FB_{vi} - (\delta C_v + E + \epsilon_{cal})F_{CAL}$$

(62)

As with chemical fluxes, substituting in expressions for concentration driven diffusion, membrane permeability to $HCO_3^-$ in terms of background utilization,

**Table 2 | Parameters from the model, their definitions, derivation and units.**

| Symbol | Definition | Derivation | Quantity | Units |
|---|---|---|---|---|
| *Measured parameters* | | | | |
| $C_e$, $B_e$ | External $CO_2$ and $HCO_3^-$ concentration | Measured [DIC] and pH | Independent variable | mol m$^{-3}$ |
| $\delta C_e$, $\delta B_e$ | $\delta^{13}$C of external $CO_2$ & $HCO_3^-$ | Calculated from $\delta^{13}$C of DIC and pH | Independent variable | ‰$_{PDB}$ |
| $F_{FIX}$ | Average carbon fixation rate | Division rate, mol C org per cell | Measured variable | mol s$^{-1}$ |
| $F_{CAL}$ | Average calcification rate | Division rate, mol calcite per cell | Measured variable | mol s$^{-1}$ |
| PIC:POC | Particulate inorganic to particulate organic carbon ratio of biomass | $F_{CAL}/F_{FIX}$ | Measured variable | Molar ratio |
| TCR | Total carbon assimilation rate | $F_{CAL} + F_{FIX}$ | Measured variable | mol s$^{-1}$ |
| $\delta C_{org}$ | $\delta^{13}$C of organic material | Measured directly | Measured variable | ‰$_{PDB}$ |
| $\delta C_{cal}$ | $\delta^{13}$C of calcite | Measured directly | Measured variable | ‰$_{PDB}$ |
| $r_{cell}$ | Cell radius | Measured directly | Measured variable | m |
| $SA_{cell}$, $V_{cell}$ | Cell surface area and volume | Functions of $r_{cell}$ | Measured variable | m$^2$ & m$^3$ |
| Rho | Cellular organic carbon density | This study, consistent with Reinfelder[77] | $20 \times 10^{-15}$ | mol m$^{-3}$ |
| *Parameters from literature* | | | | |
| $\epsilon_{cal}$ | Carbon isotopic fractionation during calcification | Zeebe and Wolf-Gladrow[44] | $+1$ (from $HCO_3^-$) | ‰$_{PDB}$ |
| $\epsilon_{cb1}$, $\epsilon_{bc1}$, $\epsilon_{bc4}$, $\epsilon_{bc1}$, $\epsilon_{cbCA}$, $\epsilon_{bcCA}$ | Kinetic carbon isotopic fractionation factors | Zeebe and Wolf-Gladrow[44] | $-13$, $-22$, $-11$, $-20$, $-1$, $-10$ | ‰$_{PDB}$ |
| $k_{p1}$, $k_{m1}$, $k_{p4}$, $k_{m4}$ | Kinetic rate constants | Calculated from Zeebe and Wolf-Gladrow[44] | T and S dependent. | various—dependent on reaction |
| $K^*_W$, | Ion product of water | Calculated from (ref. 78) | T and S dependent. | mol$^2$ m$^{-6}$ |
| $K^*_1$, | 1st acidity constant of carbonic acid | Calculated from Lueker et al.[79] | T and S dependent. | mol m$^{-3}$ |
| $\frac{K_{cat}}{K_m}$ CA | Specific activity of CA for hydration reaction | Inferred from bovine erythrocyte CA[45] | $2.7 \times 10^7$ | M$^{-1}$ s$^{-1}$ |
| $pH_i$, $pH_x$, $pH_v$ | pH in compartments i, x & v. | ref. 33 | 7.0, 7.9 & 7.1. | free scale |
| $f_x$, $f_v$ | Scale factor for compartments x & v | TEM images[80,81] | 1.1 & 0.6 | dimensionless |
| $S_x$, $S_v$ | Shape factor for compartments x & y | TEM images[80,81] | 6 & 4 | dimensionless |
| *Fitted parameters* | | | | |
| $\epsilon_{fe}$ | Effective carbon isotopic fractionation during C-fixation by Rubisco | Fitted parameter | $-14.3$ | ‰$_{PDB}$ |
| $P_{Cei}$ | Cell membrane permeability to $CO_2$ | Fitted parameter | $9.3 \times 10^{-4}$ | ms$^{-1}$ |
| $P_{Bei}$ $(T_0, T_U)$ | Cell membrane permeability to $HCO_3^-$ | Fitted parameter | $7.8 \times 10^{-8}$, $5.1 \times 10^{-6}$ | ms$^{-1}$ |
| $G$ | Effective concentration ratio of the coported ion at the cell membrane | Fitted parameter | 2.7 | ratio |
| [CA]$_i$, [CA]$_x$, [CA]$_v$ | [CA] in compartments i, x & v. | Inferred | 0.1–1 | mol m$^{-3}$ |

See supplementary information for values for measured parameters generated in this study.

$HCO_3^-$ concentration in terms of $CO_2$ and $H^+$, and rearranging gives:

$$\delta C_v = \frac{\delta C_i \frac{C_i}{C_e}\left(1 + (1+E)\frac{K^*_{H_2CO_3}}{H_i}(T_0 + T_U U_0)\right) - \frac{(E + \epsilon_{cal})F_{CAL}}{f_v P_C SAC_e}}{\frac{-F_{CAL}}{f_v P_C SAC_e} + \frac{C_v}{C_e}\left(1 + (1+E)\frac{K^*_{H_2CO_3}}{H_v}(T_0 + T_U U_0)\right)} \qquad (63)$$

$$\delta C_x = \frac{\delta C_i \frac{C_i}{C_e}\left(1 + (1+E)\frac{K^*_{H_2CO_3}}{H_i}(T_0 + T_U U_0)\right) - \frac{\epsilon_f F_{FIX}}{f_x P_C SAC_e}}{\frac{-F_{FIX}}{f_x P_C SAC_e} + \frac{C_x}{C_e}\left(1 + (1+E)\frac{K^*_{H_2CO_3}}{H_x}(T_0 + T_U U_0)\right)} \qquad (64)$$

$$\delta C_i = \frac{\delta C_e\left(1 + (1+E)G\frac{K^*_{H_2CO_3}}{H_e}(T_0 + T_U U_0)\right) - \frac{(\delta C_x + \epsilon_f)F_{FIX}}{P_C SAC_e} - \frac{(\delta C_v + E + \epsilon_{cal})F_{CAL}}{P_C SAC_e}}{\frac{C_i}{C_e}\left(1 + (1+E)\frac{K^*_{H_2CO_3}}{H_i}(T_0 + T_U U_0)\right)} \qquad (65)$$

The chemical carbon flux equations (50), (54) and (58) can be solved simultaneously. Further, the carbon isotope flux equations (63)–(65) can also be solved simultaneously, using the chemical carbon flux equations. In these expressions, including $U_0$, the rates of fixation and calcification always occur divided by $P_C SAC_e$, and this is also the only appearance of these parameters. Thus,

$$\frac{F_{CAL}}{P_C SAC_e} \quad \text{and} \quad \frac{F_{FIX}}{P_C SAC_e} \qquad (66)$$

are compound varibles that describe the entire system when constant and intracellular pH is assumed and known a priori. In this model, the fixation rate is related to cell size via the equation,

$$F_{FIX} = \rho V \mu \qquad (67)$$

where $\rho$ is the cellular carbon density and $V$ and $\mu$ are the volume and division rate of a cell respectively. As PIC:POC is equal to $\frac{F_{CAL}}{F_{FIX}}$, the compound parameters become:

$$\frac{F_{FIX}}{P_C SAC_e} = \frac{\rho V \mu}{P_C SAC_e} = \frac{r \rho \mu}{3 P_C C_e} \qquad (68)$$

and:

$$\frac{F_{CAL}}{P_C SAC_e} = \frac{PIC : POC \rho V \mu}{P_C SAC_e} = \frac{PIC : POC r \rho \mu}{3 P_C C_e}. \qquad (69)$$

Therefore, at the limit, where intracellular CA is infinitely high such that all compartments are in chemical and isotopic equilibrium, the orthogonal variables that describe the entire system are:

$$PIC : POC \quad \text{and} \quad \tau = \frac{r \rho \mu}{3 P_C C_e}. \qquad (70)$$

$\tau$ is a parameter that is similar to the ratios that evolved through early work describing carbon fluxes in phytoplankton[16–18,21,23–25,32]. The more complicated disequilibrium model collapses to a state completely consistent with this earlier work when equilibrium is assumed and PIC:POC = 0. When the cell is close to chemical and isotopic equilibrium with respect to carbon therefore, changes in cell size, growth rate and extracellular $[CO_2]$ cannot be decoupled.

**Fitted parameters.** The results from culture were used to constrain $P_{Ccell}$ (the permeability of membranes to $CO_2$), $T_0$, $T_U$ (the constants required to define the linear dependence of membrane permeability to $HCO_3^-$ on utilization (equation (4)), $G$ (the asymmetry of the effective cell membrane permeability to $HCO_3^-$, which may reflect the extra-cellular to intra-cellular concentration ratio of the coported ion), and $\epsilon_{fe}$ (the effective enzymatic isotopic fractionation associated with carbon fixation by RuBisCO). Universal constants were fitted to the data by minimization of a misfit function, which quantified the sum of squared deviations of the modelled results from the measured data. Minimization of the misfit function was achieved using an R implementation[71] of the constrained DIRECT_L algorithm[72] within a defined parameter space. The model was prescribed to be undefined when concentrations are negative, or when the saturation state of calcite in the coccolith vesicle is below 0.1. Final parameter values were checked with an alternative implementation of the model in Python, using a combination of a broad search with the Differential Evolution algorithm[73], followed by refinement of its result with the Nelder-Mead Simplex algorithm[74]. Only complete data sets were used to computationally constrain the universal parameters of the model. The required parameters are: cell size, PIC:POC, growth rate, medium pH and [DIC], $\delta^{13}C_{DIC}$, $\delta^{13}C_{calcite}$ and $\delta^{13}C_{org}$. The model was fitted to these data multiple times with pseudo-replicate datasets generated by resampling each value from an assumed gaussian distribution defined by the expected value and uncertainty associated with each measurement. All input parameters were resampled in this way. The fitted value of $P_{Ccell}$ is $\sim 9.3 \times 10^{-4}\,ms^{-1}$, which is at the high end of the range suggested by Hopkinson et al.[46] for diatoms. $T_0$ is $\sim 7.8 \times 10^{-8}\,ms^{-1}$; also consistent with values given by Hopkinson et al.[46] Based on these results, at low utilization, the membrane is 3–4 orders of magnitude less permeable to $HCO_3^-$ as to $CO_2$. The constant $T_U$ describing the increase in membrane permeability to $HCO_3^-$ with increased utilization is $\sim 5.1 \times 10^{-6}\,ms^{-1}$ per unit utilization. The parameter $G$, which describes the asymmetry of the cell membrane permeability to

$HCO_3^-$, is constrained to be 2.7. This value partially reflects the $Na^+$ gradient, but is somewhat too low, so probably also represents $Cl^-$-coupled anion exchange, which would drive bicarbonate out of the cell. When utilization is low, either at high DIC or in small cells, $\sim 10\%$ of carbon enters the cell in the form of $HCO_3^-$. When cells experience high utilization, at low DIC or in larger cells, the contribution of $HCO_3^-$ to total DIC supply can be as high as 50%. CA in all compartments is between 0.1 and 1 mM in all intracellular compartments, based on a specific activity of $2.7 \times 10^7\,M^{-1}\,s^{-1}$ (ref. 45). Values below 0.1 mM lead to extremely unstable model output and increasing this value above 0.1 has little effect on model output suggesting that compartments are approximately in equilibrium with respect to carbon. As it takes an order of magnitude longer to reach equilibrium with respect to oxygen, and we know that the cell cannot be in equilibrium with respect to oxygen due to the existence of oxygen isotopic vital effects in coccolith calcite, the CA concentration must exist within this small range. $\epsilon_{fe}$ is $\sim -14.3\%$, which is far smaller than that used in the literature to date[2,21,24,25,32] but is closer to the in vitro measured value of $\epsilon_f$ given for E. huxleyi ($-11\%$ (ref. 39)) and for the diatom Skeletonema costatum ($-19\%$ (ref. 53)). Parameters associated with the model input and output are summarized in Table 2.

**Data availability.** The culture data that support the findings of this study are contained in the Supplementary Material, and in the literature cited throughout the manuscript.

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

## Acknowledgements

This study was conducted at the University of Oxford (UK). H.L.O.M. was funded by PhD studentship NE/I019522/1 in association with UKOARP. R.E.M.R. acknowledges NERC grant NE/H017119/1 and ERC grant SP2-GA-2008-200915. The work of J.B. was funded by NERC National Capability in Marine Modelling. We thank Ian Probert for providing strains. We thank P. Ziveri and two other reviewers for their valuable comments on this manuscript.

## Author contributions

H.L.O.M. and J.B. wrote the model. H.L.O.M. and M.H. did the culture experiments. H.L.O.M. interpreted the results and wrote the paper in discussion with R.E.M.R., M.H. and J.B.

## Additional information

**Competing financial interests:** The authors declare no competing financial interests.

