## [Peer Review File · Nature Communications]

Reviewers' comments:

Reviewer #1 (Remarks to the Author):

This is a very interesting, detailed, clearly written and useful manuscript that presents a model based on generally well-supported assumptions for inorganic carbon transport and utilization in coccolithophores to understand the mechanistic basis of carbon isotope fractionation in coccolith calcite, including the distinction between isotopically "heavy" and "light" species. The model is constrained by culture experiments with three species of coccolithophore representing fast and slow growers. Subject to consideration of the comments below, this should be an extremely valuable contribution to understanding the interpretation of palaeoproxies and should significantly increase the understanding of the mechanistic interplay between photosynthetic and calcification carbon fluxes. It is likely to be of interest to a wide readership.

I do have a couple of questions for clarification. My comments mainly relate to the basic biological assumptions rather than the construction of the model itself.

1. Figure 2. The legend was initially not clear to me. Perhaps it would be better to say "model output with data superimposed". I do have an issue with the statement about *E. huxleyi* $\delta^{13}\text{C}_{\text{cal}}$ increasing towards low DIC before sharply decreasing at very low values of DIC. This sharp decline is not clear from the figure. It would be better to expand the DIC scale at these low values to make this clear. Moreover, there is not any supporting data for these low DIC levels and it has been shown by a number of studies that calcification is inhibited at DIC values generally lower than 0.5 mM. How then, can the model predict any values for calcite $\delta^{13}\text{C}$ at low DIC?

2. I would question the assumption (lines 407-415) that HCO_3^- movement across membranes is down-gradient and that Cl^- or Na^+ concentration gradients are negligible. In seawater the concentration gradients for both of these ions across the plasma membrane are substantial. While the Cl^- electrochemical potential difference may be lower due to the negative membrane potential, there will be a very large inward electrochemical potential gradient for Na^+ . This gradient could well drive active HCO_3^- uptake and accumulation against a concentration gradient. Indeed there is evidence for active HCO_3^- accumulation operating in carbon concentrating mechanisms. How would the model behave if HCO_3^- uptake was driven uphill in this way?

3. I am intrigued by the statement (line 469-471) that if future work concludes that external CA is absent in one or more species, it may be necessary to allow chemical and isotopic disequilibrium in the boundary layer. This begs the question of how the model will behave if the speciation of carbon in the boundary layer is significantly limited in this way. At least some speculation or model simulation would be useful in this context.

4. Minor point: The References are not presented in a consistent format. For example some have all authors listed, some only the first three and some have et al.

Reviewer #2 (Remarks to the Author):

Isotopic fractionation in biologically precipitated calcite or 'vital effects' is a very fascinating and original topic enhancing the understanding of biomineralization, physiology and biogeosciences. In marine calcifying organisms present as fossils in the sedimentary record, this understanding can improve the application of geochemical isotopic proxies for paleoceanographic and paleoclimate reconstructions. Although the awareness about vital effects has been available for a few decades, there is still little work done on this topic.

Coccolithophores as calcifying marine phytoplankton are a particularly interesting group of calcifiers to study since they have a very long and continuous fossil record, are major carbonate producers, and some species like the cosmopolitan and abundant *Emiliana huxleyi* are largely studied in laboratory for physiological and chemical studies. In particular, they are known for their large oxygen and carbon vital effects in their coccolith calcite making them critical for understating biological fractionation.

The paper by McClelland et al. is focusing on the carbon isotope vital effects in coccolith calcite and proposes a mechanistic understanding of the large species-specific isotopic fractionation. The key results relate to the fact that carbon (and oxygen) isotopic fractionation is largest when cellular carbon utilization is high, PIC/POC is low, and disappears when PIC/POC is high and carbon utilization is low. When the cellular carbon requirement is an appreciable fraction of the carbon pool to the cell, vital effects are largest. The main basic idea is that the isotopic composition of coccolith calcite is a function of the carbon fixation by photosynthesis and the calcification process having opposite effects on the isotopic composition of the coccolith vesicle bicarbonate pool.

The effect of PIC/POC changes on the carbon isotopic composition of the organic matter is weak because the effect of calcification on the CO₂ isotopic composition in the chloroplast is far outweighed by that of carbon fixation. The proposed model is based on a competing Rayleigh-type fractionation process describing the partitioning of isotopes between two reservoirs. In the paper this is used within the cell affecting the bicarbonate pool in the intracellular coccolith vesicle where calcification of coccoliths is occurring. The kinetic isotopic fractionations of biologically-mediated processes can vary in magnitude, depending on several factors and in the case of coccolithophores includes species-specific metabolic transformations.

The authors need to elaborate on the following statements:

lines 113-134: HCO₃ as well as CO₂ usage. This is partially correct since it doesn't seem to be clear that 100% CO₂ can never work.

lines 134-136: Published values for H⁺ concentration in different compartment. These values are published but are controversial since it's technically impossible to measure. The cytosolic pH is probably correct.

lines 150-155: 'Carbonate system reactions are quantifiable functions of intracompartamental DIC-species-specific concentrations, H⁺ concentration (pH), and the concentration of carbonic anhydrase (CA), the metalloenzyme responsible for catalysing the otherwise sluggish interconversion of CO₂ and HCO₃⁻.' This is not clear. Clarify about the quantification. Where do all these come from? Ref? Ad-hoc assumption?

The model assumptions are in general reasonable, however several in-depth clarifications are needed. In lines 412-415 it is stated: transmembrane concentration gradient for Cl⁻ and Na⁺ is negligible. This is highly unlikely as shown for example in the work by Taylor, 2009 on diatoms, which demonstrated of voltage-activated Na⁺ channels and the capacity to generate fast Na⁺-based action potentials in a unicellular photosynthetic organism and in *Coccolithus pelagicus* on the Cl⁻-channel by Taylor and Brownley, 2003 proposing that 'major ion conductances play an essential role in membrane voltage regulation that relates to the unique transport physiology of these calcifying phytoplankton'. In addition, the concentration gradient of ions does not determine their trans-membrane flux, but the electro-chemical gradient does, i.e. one has to know the membrane voltage too.

The assumption that 'CA is present in high enough concentrations that the external surface of the cell is in chemical and isotopic equilibrium with the bulk seawater' (lines 461-470) seems decided ad-hoc to justify the model since there is no evidence that this is the case. It is important to elaborate on the other assumption on a model where CA is absent in one or more species and

allows chemical and isotopic disequilibrium in the cellular boundary layer.

It is interesting the comparison with oxygen isotopic fractionation in coccolith calcite when compared with carbon but the discussion provided is not very convincing (lines 658-676) and why the a plot showing the $\delta^{13}\text{C}/\delta^{18}\text{O}$ is not provided? A better elaboration of the interpretation of oxygen versus carbon isotopic fractionation should be provided and compared also with the model presented for *Calcidiscus leptoporus* oxygen fractionation in Ziveri et al., 2012.

In *Calcidiscus leptoporus* and several planktic foraminifera species the calcite $\delta^{18}\text{O}$ is dependent on the seawater carbonate chemistry in addition to its dependence on temperature. The conceptual model based on the contribution of $\delta^{18}\text{O}$ -enriched HCO_3^- to the CO_3^{2-} pool in the calcifying vesicle, could explain the $[\text{CO}_3^{2-}]$ effect on $\delta^{18}\text{O}$ for the different unicellular calcifiers (Ziveri et al., 2012). Please elaborate on this since the model do not fit with the proposed in this work. Also please consider that there are evidence that calcification in *C. leptoporus* may occur in a highly alkaline environment (Gussone et al., 2007; Ziveri et al., 2012; Hermoso et al., 2014).

Since this paper aims to show the evolution of the carbon isotope vital effects in coccolith calcite, why were experiments on only *E. huxleyi* and *Gephyrocapsa oceanica* performed? Why not species with high PIC/POC (e.g. *C. leptoporus*, *H. carteri*)? And why 'evolution' when is proposing a mechanistic understanding?

I found the last part of the paper on coccolith-based geochemical paleoclimate proxy for pCO_2 atm very speculative. It is known that the $\delta^{13}\text{C}$ in coccolithophores-producing alkenones is very limiting but in the paper I didn't find any robust suggestion for improving it. A robust proxy for coccolithophore PIC/POC for paleoreconstruction is not available at this moment.

Finally, I found the paper interesting and possibly of interest for a publication in Nature. However, some major improvement needs to be done on the model assumptions, carbon versus oxygen vital effects and paleoclimate proxy discussion.

Reference:

Hermoso, 2014, *Cryptogamie Algologie* 35(4):323-352.

Taylor A, 2009, PlosONE
(<http://dx.doi.org/10.1371/journal.pone.0004966>),

Taylor A and Brownley C, 2003, *Plant Physiol.* 2003 Mar; 131(3): 1391-1400, doi:
10.1104/pp.011791.

Ziveri P, Thoms S, Probert I, Geisen M, Langer G. (2012) *Biogeosciences*, 9, 1-8, 2012.

Reviewer #3 (Remarks to the Author):

The manuscript "The origin of carbon isotope vital effects in coccolith calcite" by HLO McClelland et al. presents an interesting approach to describe and quantify cellular C budgets in coccolithophores.

The authors developed a detailed model, which is a great step towards a better understanding of the carbon budget of coccolithophores, important unicellular marine algae, which significantly

contribute to the marine primary production. Nevertheless, there are two main issues related to the presented model and proposed proxy application.

Main points of criticism:

The model appears to be biologically solid, but some of the model predictions are not apparent for the measured d13C data. For instance the high d13C at low DIC (Fig. 2) predicted from the model is not reproduced by the analysed coccolith of *E. huxleyi*. Similarly, the increase in d13C towards low DIC values is also hardly seen in the experimental Corg data. Therefore, the model data is clearly within uncertainty consistent with the experimental data, but the experimental data seem to provide no independent evidence for some features of the d13C vs DIC curves. Consequently, the model output (Fig.3) is somewhat questionable.

While the significance of coccolithophores for the global C cycle is undoubted, their potential as geochemical proxy archive and the application of the presented approach is restricted by problems related to the isolation of pure mono-specific fractions of coccolith calcite from sediments and compound specific C isotope analyses of molecules that origin exclusively from a single coccolithophore species. To demonstrate the general applicability of the approach and increase the significance of the study, it would be desirable to provide d13C data of different coccolith and compound specific d13C org of coretop sediments, to illustrate that at least in modern sediments the methods yields consistent data.

Detailed comments:

The statement in lines 28/29: "..., which is consistent with oxygen isotopes" is misleading, as later in the text (line 650) it is stated that "...this model cannot currently be fully resolved with respect to oxygen isotopes"

Figure 1: subscripts are quite small, please consider to increase font size.

Lines 218 ff and Figure 2: indicate in Figure 2 or its caption, which coccolithophores represent high or low PIC/POC, respectively

Figure S2: for consistency "Mu" should read " μ "

Line 2681: please specify which figure ("Fig. ??") you intended to refer to

Response to reviewer comments

McClelland et al.

We would firstly like to thank all three reviewers for their careful and insightful comments, and for their enthusiasm in our work. We are pleased to have the opportunity to address their concerns and improve this manuscript. There were some common points among these reviews which we have grouped and responded to together. The more specific points raised by the reviewers are individually addressed throughout the following document. Our responses are in blue sans-serif font below each comment.

Main points

1. Data in Figure 2

- R.#1** Figure 2. The legend was initially not clear to me. Perhaps it would be better to say "model output with data superimposed". I do have an issue with the statement about *E. huxleyi* $\delta^{13}\text{C}_{\text{cal}}$ increasing towards low DIC before sharply decreasing at very low values of DIC. This sharp decline is not clear from the figure. It would be better to expand the DIC scale at these low values to make this clear. Moreover, there is not any supporting data for these low DIC levels and it has been shown by a number of studies that calcification is inhibited at DIC values generally lower than 0.5 mM. How then, can the model predict any values for calcite $\delta^{13}\text{C}$ at low DIC?
- R.#3** The model appears to be biologically solid, but some of the model predictions are not apparent for the measured $\delta^{13}\text{C}$ data. For instance the high $\delta^{13}\text{C}$ at low DIC (Fig. 2) predicted from the model is not reproduced by the analysed coccolith of *E. huxleyi*. Similarly, the increase in $\delta^{13}\text{C}$ towards low DIC values is also hardly seen in the experimental Corg data. Therefore, the model data is clearly within uncertainty consistent with the experimental data, but the experimental data seem to provide no independent evidence for some features of the $\delta^{13}\text{C}$ vs DIC curves. Consequently, the model output (Fig.3) is somewhat questionable.

To address the concerns of the reviewers, we have now compiled and added all comparable data from the literature to Figure 2 [10, 2], and we have completely reworded the figure

caption to address the comments of Reviewer 1, and included their suggestions. We have also changed the X-axis to CO₂ concentration rather than DIC for consistency with the literature. As can be seen in our new Figure 2, the model successfully predicts the trends seen in these newly incorporated data. At high CO₂, calcite carbon isotopic compositions converge to abiogenic equilibrium calcite compositions (horizontal dotted line). At low CO₂ these compositions diverge (towards the left of the plot), trending towards heavy values in the low PIC:POC coccolithophore, *Pleurochrysis placolithoides*, and towards lighter values in the high PIC:POC coccolithophore *Calcidiscus leptoporus*. The slight increase in carbon isotopic compositions of *Emiliana huxleyi* calcite with decreasing CO₂ is indeed subtle ($\sim +1\%$), but it is present in our data set and also in that of Hermoso et al. (2016) [2] (also plotted).

Figure 2 shows the theoretically possible model output (i.e. where negative concentrations are not required) over CO₂ values ranging from 0 to 80 μ M. Reviewer 1 is correct that some of the included species have not yet been cultured at very low CO₂, or have been shown to be non-viable under such conditions. Thus, model predictions at very low CO₂ may not be verifiable in practice. However, we feel the limited experimental results do not warrant omission of the low CO₂ space as non-viable. We have therefore chosen to show model results for the entire CO₂ range, while now noting in the caption that parts of this model space are currently hypothetical in nature.

The rapid return to light values for the low PIC:POC coccolithophore species at extreme low CO₂ mentioned by Reviewer 1 is a description of the mathematics and is not theoretically realizable in most organisms. However, we feel it is important to make this point, because the curves are all described by the same set of equations, and critically, there is not a fundamental discontinuation between the trends towards heavy and light vital effects. The opposing behavior of the heavy and light groups is simply due to the relative magnitude of competing processes. Calcite carbon isotopic compositions are predicted to return to light values for coccolith calcite across all species - it just happens that this part of the curve exists in a realizable parameter space in the high PIC:POC species, but not in the low PIC:POC species. The figure caption has been reworded to reflect this.

For the organic carbon isotopic data, we have included the results from dilute batch culture experiments on *E. huxleyi* by Riebesell et al. (2000) [10], which show a strong increase in carbon isotopic compositions with decreasing CO₂, at values of CO₂ lower than our dataset. These data overlay our data very well, and clearly show the predicted increase in isotopic composition of *E. huxleyi* calcite at very low CO₂.

We have additionally provided a supplemental figure that includes C_{org} data from diatoms for reference as an end-member scenario for a closely related organism where PIC:POC = 0. We included all of the data from the literature we could find for dilute batch culture experiments on diatoms. The consistency of these data with our model is a testament to the robustness of the model. Their inclusion is appropriate because these are the data upon which the original classic description of the trends in organic carbon isotopic compositions are based, and onto which our model collapses at intracellular equilibrium and PIC:POC = 0. This supplemental figure also includes data from Rickaby et al. (2010) [9] whose calcite carbon isotopic compositions in

G. oceanica and in *C. pelagicus* perfectly overlay the others presented here. The organic carbon isotopic compositions in this study are questionable however, particularly those of *G. oceanica*, which are inconsistent with all other datasets in the literature, which unanimously describe an increasing magnitude of isotopic fractionation of carbon into organic matter with increasing CO₂. We therefore have included these data in the supplement for completion, but do not present them in the figure in the main manuscript. We have explained the omission of these data in the figure caption of Fig. S5.

The additional datasets taken from literature were not used for model calibration; indeed, they cannot be used for this purpose as they lack information on some of the environmental and physiological parameters needed to drive the model. The fact that these additional datasets nevertheless overlay model curves in Fig 2/S5, both within and beyond the range of data used for calibration, demonstrates the robustness of the model.

2. Bicarbonate transport

- R.#1** I would question the assumption (lines 407-415) that HCO₃⁻ movement across membranes is down-gradient and that Cl⁻ or Na⁺ concentration gradients are negligible. In seawater the concentration gradients for both of these ions across the plasma membrane are substantial. While the Cl⁻ electrochemical potential difference may be lower due to the negative membrane potential, there will be a very large inward electrochemical potential gradient for Na⁺. This gradient could well drive active HCO₃⁻ uptake and accumulation against a concentration gradient. Indeed there is evidence for active HCO₃⁻ accumulation operating in carbon concentrating mechanisms. How would be model behave if HCO₃⁻ uptake was driven uphill in this way?
- R.#2** The model assumptions are in general reasonable, however several in-depth clarifications are needed. In lines 412-415 it is stated: transmembrane concentration gradient for Cl⁻ and Na⁺ is negligible. This is highly unlikely as shown for example in the work by Taylor, 2009 on diatoms, which demonstrated of voltage-activated Na⁺ channels and the capacity to generate fast Na⁺-based action potentials in a unicellular photosynthetic organism and in *Coccolithus pelagicus* on the Cl-channel by Taylor and Brownley, 2003 proposing that 'major ion conductances play an essential role in membrane voltage regulation that relates to the unique transport physiology of these calcifying phytoplankton'. In addition, the concentration gradient of ions does not determine their trans-membrane flux, but the electro- chemical gradient does, i.e. one has to know the membrane voltage too.

We have changed the way that bicarbonate fluxes are formulated in this model, to address these points.

Reviewer 2's point that membrane voltage (or more specifically reversal potential) determines the direction and rate of the trans-membrane flux, and equilibrium concentrations, of charged ionic species in cells relative to their environment is true when the chemical species

in question traverses the membrane in its charged form. However, in coccolithophores, the current consensus in the literature is that the SLC4 family of bicarbonate transporter proteins is responsible for bicarbonate uptake [8, 7, 12]. In bicarbonate transport facilitated by the types of proteins that are probably present in coccolithophores, the HCO_3^- ion moves across a membrane paired to another ion with a favorable trans-membrane gradient. This can be a Na^+ ion moving in the same direction (symport), or a Cl^- ion moving in the opposite direction (anti-port), such that there is no net movement of charge [11]. As a result, the force driving the combined transport of HCO_3^- and its paired ion is dependent only on the trans-membrane gradients in both ions; it is independent of the cell membrane potential. In the revised model, this is described with a formulation based on the law of mass action.

The flux of bicarbonate symported across a membrane by electroneutral transporters is therefore proportional to the product of the concentrations of the bicarbonate ion and the Na^+ ion on the proximal side of the membrane. Similarly, the flux of bicarbonate anti-ported across a membrane is proportional to the product of the concentrations of the bicarbonate ion on the proximal side of the membrane and the Cl^- ion on the distal side of the membrane. As both Na^+ and Cl^- are more abundant outside the cell than in the cytosol, Cl^- -antiporting would inhibit the influx of HCO_3^- rather than enhance it. Therefore, we assume Na^+ -symporting to be the dominant means of bicarbonate entering the cell, as suggested by Reviewer 1. We factor this into the model very simply by introducing another universal constant (G), which represents the ratio of concentrations of Na^+ outside the cell to inside the cell; this is incorporated in the model as different effective permeabilities of the cellular membrane in each direction to bicarbonate. This new constant affects only the flux of bicarbonate from the external environment to the cytosol. This is, of course, a simplification, but we feel it is the most sophisticated and realistic description of bicarbonate transport that is reasonable, when trying to extract meaning from a model. We also effectively assume that gradients in Na^+ and Cl^- between intracellular compartments are small. We explain this new formulation throughout the main manuscript and in the supplement wherever bicarbonate transport is discussed. This new formulation allows for (but does not prescribe) net up-gradient bicarbonate transport into the cell.

3. CA in the external boundary layer

R.#1 I am intrigued by the statement (line 469-471) that if future work concludes that external CA is absent in one or more species, it may be necessary to allow chemical and isotopic disequilibrium in the boundary layer. This begs the question of how the model will behave if the speciation of carbon in the boundary layer is significantly limited in this way. At least some speculation or model simulation would be useful in this context.

R.#2 The assumption that 'CA is present in high enough concentrations that the external surface of the cell is in chemical and isotopic equilibrium with the bulk seawater' (lines 461-470) seems decided ad-hoc to justify the model since there is no evidence

that this is the case. It is important to elaborate on the other assumption on a model where CA is absent in one or more species and allows chemical and isotopic disequilibrium in the cellular boundary layer.

We would like to direct the reviewers to our discussion of this phenomenon in the supplementary material (starting from line 1472). Figure S3 explores the scenario that coccolithophores do not produce extracellular CA, where the boundary layer may be altered due to the sluggish reacto-diffusive kinetics that replenish the CO_2 pool. The concentration of HCO_3^- in the boundary layer will not change as it is in great abundance, but the boundary layer can become depleted in CO_2 . In this exploration, we have assumed that there is no isotopic fractionation by membranes, so the depletion of CO_2 does not cause a Rayleigh-type fractionation within this particular pool. When external CA is not present, and when growth rates are high, cells are large, and CO_2 concentrations are low, the boundary layer becomes significantly depleted in CO_2 . Trans-membrane fluxes of carbon species are determined by the concentration at the membrane, and thus the cell "sees" only this depleted environment, which is reflected in the isotopic compositions. As shown in Figure S3.F, the greatest theoretical depletion with no external CA could lead to ambient CO_2 being underestimated by as much as 50%.

The presence of external CA is not an ad-hoc assumption - In main manuscript (lines 496-536) and the supplementary section (lines 1574-1581), we cite a number of studies that have investigated extracellular CA in coccolithophores and concluded that the presence of external CA is likely. In all other modeling work to date [6, 1, 3, 4], equilibrium of the cell's surface with the ambient water is implicitly assumed - we make the same assumption, but state that the presence of external CA is the only reasonable mechanism whereby equilibrium with the ambient water may be maintained.

4. Proposed proxy approach

- R.#2** I found the last part of the paper on coccolith-based geochemical paleoclimate proxy for pCO_2 atm very speculative. It is known that the $\delta^{13}\text{C}$ in coccolithophores-producing alkenones is very limiting but in the paper I didn't find any robust suggestion for improving it. A robust proxy for coccolithophore PIC/POC for paleoreconstruction is not available at this moment.
- R.#3** While the significance of coccolithophores for the global C cycle is undoubted, their potential as geochemical proxy archive and the application of the presented approach is restricted by problems related to the isolation of pure mono-specific fractions of coccolith calcite from sediments and compound specific C isotope analyses of molecules that origin exclusively from a single coccolithophore species. To demonstrate the general applicability of the approach and increase the significance of the study, it would be desirable to provide $\delta^{13}\text{C}$ data of different coccolith and compound specific $\delta^{13}\text{C}$ org of coretop sediments, to illustrate that at least in modern sediments the methods yields consistent data.

A direct implication of the increased understanding we have gained about carbon isotopes in coccolithophores is in the theoretical development of a CO₂ proxy - we therefore disagree with Reviewer #2 that the proxy approach we propose does not constitute a significant theoretical improvement in the alkenone CO₂ proxy. The alkenone CO₂ proxy as used in numerous high profile paleoclimate publications, relies on an assumption that the sole carbon source for coccolithophore cells is CO₂, acquired by passive diffusion. It also assumes that a coccolithophore cell is a single compartment, and it completely ignores calcite - the other preserved record of carbon isotopic compositions. At present, the alkenone CO₂ proxy requires the isotopic composition of contemporaneous seawater to be inferred from foraminifera or fish teeth. By contrast, the model presented here formally constrains the influence of CO₂ on the difference between the isotopic compositions of organic matter in large and small coccolithophores, calcite in large and small coccolithophores, and the difference between organic matter and calcite in a given species. This would potentially allow an external reference to be circumvented. Our model is of greater sophistication than the empirical relationship currently employed in the alkenone proxy, incorporating multiple intracellular compartments as well as HCO₃⁻ usage.

The purpose of this last section of the paper is to present a new concept for constraining past CO₂ concentrations, based on the wealth of new understanding that our laboratory data and model provide. We feel that a core-top calibration of this theoretical approach is beyond the scope of the current study. With respect to a possible proxy for PIC:POC mentioned by Reviewer 2, we have just had a manuscript accepted into *Scientific Reports* describing a new approach for reconstructing PIC:POC in these organisms from their ancient remains ("Calcification response of a key phytoplankton family to millennial-scale environmental change.", manuscript no. SREP-16-06353A). Further to this, obtaining compound-specific C_{org} isotopic compositions from monospecific fractions is absolutely possible, and we have done it. We are excited that Reviewer #3 pointed to these particular components of the problem. We have just had a paper accepted in *Nature Communications* describing the protocol for the extraction of polysaccharides from within the calcite lattice ("The uronic acid content of coccolith-associated polysaccharides provides an insight into coccolithogenesis and past climate", manuscript no. NCOMMS-15-16583). [Redacted]

Reviewer #1

This is a very interesting, detailed, clearly written and useful manuscript that presents a model based on generally well-supported assumptions for inorganic carbon transport and utilization in coccolithophores to understand the mechanistic basis of carbon isotope fractionation in coccolith calcite, including the distinction between isotopically "heavy" and "light" species. The model is constrained by culture experiments with three species of coccolithophore representing fast and slow growers. Subject to consideration of the comments below, this should be an extremely valuable contribution to understanding the interpretation of palaeoproxies and should significantly increase the understanding of the mechanistic interplay between photosynthetic and calcification carbon fluxes. It is likely to be of interest to a wide readership. I do have a couple of questions for clarification. My comments mainly relate to the basic biological assumptions rather than the construction of the model itself.

1. (Copied to Main Point 1.)
2. (Copied to Main Point 2.)
3. (Copied to Main Point 3.)
4. **Minor point:** The References are not presented in a consistent format. For example some have all authors listed, some only the first three and some have et al.

Response: Corrected. We have changed the format of references to follow Nature's guidelines for authors, specifically that: "All authors should be included in reference lists unless there are more than five, in which case only the first author should be given, followed by 'et al..'"

Reviewer #2

Isotopic fractionation in biologically precipitated calcite or 'vital effects' is a very fascinating and original topic enhancing the understanding of biomineralization, physiology and biogeosciences. In marine calcifying organisms present as fossils in the sedimentary record, this understanding can improve the application of geochemical isotopic proxies for paleoceanographic and paleoclimate reconstructions. Although the awareness about vital effects has been available for a few decades, there is still little work done on this topic. Coccolithophores as calcifying marine phytoplankton are a particularly interesting group of calcifiers to study since they have a very long and continuous fossil record, are major carbonate producers, and some species like the cosmopolitan and abundant *Emiliania huxleyi* are largely studied in laboratory for physiological and chemical studies. In particular, they are known for their large oxygen and carbon vital effects in their coccolith calcite making them critical for understating biological fractionation. The paper by McClelland et al. is focusing on the carbon isotope vital effects in coccolith calcite and proposes a mechanistic understanding of the large species-specific isotopic fractionation. The key results relate to the fact that carbon (and oxygen) isotopic fractionation is largest when cellular carbon utilization is high, PIC/POC is low, and disappears when PIC/POC is high and carbon utilization is low. When the cellular carbon requirement is an appreciable fraction of the carbon pool to the cell, vital effects are largest. The main basic idea is that the isotopic composition of coccolith calcite is a function of the carbon fixation by photosynthesis and the calcification process having opposite effects on the isotopic composition of the coccolith vesicle bicarbonate pool. The effect of PIC/POC changes on the carbon isotopic composition of the organic matter is weak because the effect of calcification on the CO₂ isotopic composition in the chloroplast is far outweighed by that of carbon fixation. The proposed model is based on a competing Rayleigh-type fractionation process describing the partitioning of isotopes between two reservoirs. In the paper this is used within the cell affecting the bicarbonate pool in the intracellular coccolith vesicle where calcification of coccoliths is occurring. The kinetic isotopic fractionations of biologically-mediated processes can vary in magnitude, depending on several factors and in the case of coccolithophores includes species-specific metabolic transformations. The authors need to elaborate on the following statements:

- **lines 113-134:** HCO₃ as well as CO₂ usage. This is partially correct since it doesn't seem to be clear that 100% CO₂ can never work.

Response: We agree that it is not known that 100% CO₂ can never work - this is why we allow the model to constrain parameters that quantify the bicarbonate uptake required to fit the data - with the option of zero bicarbonate uptake explicitly allowed (See section S1). We have additionally reemphasized throughout the manuscript that the model can collapse back onto purely CO₂ supply by passive diffusion, if warranted by observations used to constrain the universal constants.

- **lines 134-136:** Published values for H⁺ concentration in different compartment. These values are published but are controversial since it's technically impossible to measure. The cytosolic pH is probably correct.

Response: We understand that these values are controversial, however, the values of Anning et al. are the only widely cited empirical (i.e. model-independent) values available in the literature for the coccolith vesicle and the chloroplast in multiple species. They are based on an established protocol, whereby cells were loaded with a pH-dependent fluorescent dye prior to analysis with a confocal laser scanning microscope. The model is able to reproduce the same curves if other hypothetical pH combinations are prescribed, however, we are reluctant to present results that are based on non-empirical input.

- **lines 150-155:** 'Carbonate system reactions are quantifiable functions of intracompartamental DIC-species-specific concentrations, H⁺ concentration (pH), and the concentration of carbonic anhydrase (CA), the metalloenzyme responsible for catalysing the otherwise sluggish interconversion of CO₂ and HCO₃.' This is not clear. Clarify about the quantification. Where do all these come from? Ref? Ad-hoc assumption?

Response: The intention of this sentence is merely to list the interacting components of the equations that describe carbonate chemistry in any given pool. We have now referenced the widely used book by Zeebe and Wolf-Gladrow, and two more recent papers that discuss CA in depth. Other than pH and CA, all of these parameters are direct outputs of the model.

The value for intracellular CA was initially prescribed to be 0.1mM, which is close to that inferred by Hopkinson et al. (2011) for diatoms [5] - the model output shows that values of CA below this cause highly unrealistic model behavior no matter what parameters are used, and that concentrations above this have a fairly small effect on the model output. We therefore infer that the carbonate chemistry in the cell is close to equilibrium with respect to carbon. It cannot be in equilibrium with Oxygen (which takes 10x longer) because oxygen isotopic vital effects would not exist, and would instead simply reflect that of carbonate chemistry in equilibrium with water. There is therefore a fairly narrow range of CA (~ 0.1mM to ~ 1mM) that makes sense. We have amended the supplementary discussion to make these conclusions about CA clearer (line 2740-).

- (Copied to Main Point 2.)
- (Copied to Main Point 3.)
- It is interesting the comparison with oxygen isotopic fractionation in coccolith calcite when compared with carbon but the discussion provided is not very convincing (lines 658-676) and why the a plot showing the d13C/d18O is not provided? A better elaboration of the interpretation of oxygen versus carbon isotopic fractionation should be provided and compared also with the model presented for *Calcidiscus leptoporus* oxygen fractionation in Ziveri et al., 2012.

In *Calcidiscus leptoporus* and several planktic foraminifera species the calcite $\delta^{18}\text{O}$ is dependent on the seawater carbonate chemistry in addition to its dependence on temperature. The conceptual model based on the contribution of ^{18}O -enriched HCO_3^- to the CO_3^{2-} pool in the calcifying vesicle, could explain the $[\text{CO}_3^{2-}]$ effect on $\delta^{18}\text{O}$ for the different unicellular calcifiers (Ziveri et al., 2012). Please elaborate on this since the model do not fit with the proposed in this work. Also please consider that there are evidence that calcification in *C. leptoporus* may occur in a highly alkaline environment (Gussone et al., 2007; Ziveri et al., 2012; Hermoso et al., 2014).

Response: It was an oversight on our part not to include a plot of oxygen isotopic compositions of calcite when preparing the manuscript. These data are now included (Fig S6), along with some aspects of the model output that informed our discussion of oxygen isotopes.

Whilst we agree that the oxygen isotopic composition of calcite is also a function of carbonate chemistry in addition to temperature, and indeed explain this in the discussion (lines 746-776), the purpose of this paper is exclusively to describe carbon isotopes. The mechanistic understanding behind carbon isotopic vital effects that the model provides is highly informative, but because the kinetic isotopic fractionation factors (KIFs) are not known for oxygen, for the hydration and dehydration of CO_2 and HCO_3^- respectively, oxygen isotopes cannot be modeled explicitly like carbon can. Any arguments for the origin of oxygen isotopic vital effects are therefore empirical and 'arm wavy' at best. We do not directly dispute the model of Ziveri et al. 2012, but also find their conclusions to be circumstantial. In particular, they propose the oxygen isotopic composition of the carbonate pool in the coccolith vesicle to be set by the ratio of HCO_3^- to CO_3^{2-} entering the coccolith vesicle. We do not agree that this ratio sets the isotopic composition of oxygen in the carbonate system of the coccolith vesicle because CO_2 forms such a large component of the transmembrane flux of DIC. The best we can do is to discuss whether oxygen isotopes are consistent with the model inferred fluxes that explain the carbon isotopic data. We feel that a cross-plot of oxygen vs. carbon isotopic vital effects would be of limited value.

We have now included an additional supplementary figure (S6) that shows the aspects of the model output that informed our qualitative discussion of oxygen. The pattern in oxygen isotopes are qualitatively the same as for carbon: as can be seen in this figure, *E. huxleyi* precipitates the heaviest calcite and *C. pelagicus* the lightest; and with the vital effects being largest at low CO_2 . However, because HCO_3^- is heavier than CO_2 in carbon but is lighter in oxygen, these trends in vital effects absolutely cannot be simultaneously explained by a change in the ratio of bicarbonate to CO_2 entering the cell or the coccolith vesicle. Similarly, one of the major sources of carbon isotopic vital effects that we discovered - the Rayleigh-type distillation within the chloroplast which causes a leakage of heavy carbon, is only relevant for carbon and is independent of oxygen. The up-regulation of HCO_3^- uptake at high cellular utilization results in larger cells taking up more HCO_3^- at a given CO_2 concentration, consistent with the

smaller forms precipitating heavier calcite than the larger forms. We hypothesize that the universal drift towards equilibrium values at high CO₂ in oxygen is due to a faster rate of hydration and dehydration (for each hydration/dehydration cycle, the oxygen in the carbonate system becomes iteratively replaced by oxygen in water) - as shown in Fig S5.C.

- Since this paper aims to show the evolution of the carbon isotope vital effects in coccolith calcite, why were experiments on only *E. huxleyi* and *Gephyrocapsa oceanica* performed? Why not species with high PIC/POC (e.g. *C. leptoporus*, *H. carteri*)? And why 'evolution' when is proposing a mechanistic understanding?

Response: The word "origin" in the title does not refer to an evolutionary origin, but to a mechanistic origin. We like the title as it is, but if preferred by the editor, could change this to "source". Regarding other species with high and low PIC:POC please see our response to Main Point 1.

- (Copied to main point 4.)
- Finally, I found the paper interesting and possibly of interest for a publication in Nature. However, some major improvement needs to be done on the model assumptions, carbon versus oxygen vital effects and paleoclimate proxy discussion.

Reference:

Hermoso, 2014, *Cryptogamie Algologie* 35(4):323-352.
Taylor A, 2009, PlosONE (<http://dx.doi.org/10.1371/journal.pone.0004966>),
Taylor A and Brownley C, 2003, *Plant Physiol.* 2003 Mar; 131(3): 1391-1400,
Ziveri P et al. (2012) *Biogeosciences*, 9, 1-8, 2012.

Reviewer #3

The manuscript "The origin of carbon isotope vital effects in coccolith calcite" by HLO McClelland et al. presents an interesting approach to describe and quantify cellular C budgets in coccolithophores.

The authors developed a detailed model, which is a great step towards a better understanding of the carbon budget of coccolithophores, important unicellular marine algae, which significantly contribute to the marine primary production. Nevertheless, there are two main issues related to the presented model and proposed proxy application.

Main points of criticism:

- (Copied to main point 1.)
- (Copied to main point 4.)

Detailed comments:

- The statement in lines 28/29: "..., which is consistent with oxygen isotopes" is misleading, as later in the text (line 650) it is stated that "...this model cannot currently be fully resolved with respect to oxygen isotopes"

Response: The compatibility of our model with trends seen in oxygen isotopes, as discussed in the text is important because contrasting theories - such as vital effects being driven purely by bicarbonate fluxes - are directly contradicted by these trends. We have changed this wording to "..., which is compatible with trends in oxygen isotopes", which we hope makes our meaning clearer. Due to the large interest in oxygen isotopes in the paleoclimate community, we feel it is important to reference oxygen, as the conclusions presented here have highly relevant implications - through cellular fluxes of carbon species - even though a formal modeling of oxygen isotopes is not possible.

- Figure 1: subscripts are quite small, please consider to increase font size.

Response: We have increased the font size of the subscripts by two sizes.

- Lines 218 ff and Figure 2: indicate in Figure 2 or its caption, which coccolithophores represent high or low PIC/POC,

Response: We have now included this in the figure legend.

- respectively Figure S2: for consistency "Mu" should read " μ "

Response: Changed.

- Line 2681: please specify which figure ("Fig. ??") you intended to refer to

Response: Corrected.

References

- [1] Bolton CT and Stoll HM (2013): *Nature*, 500(7464):558–62.
- [2] Hermoso M, Chan IZX, McClelland HLO, Heureux aMC and Rickaby REM (2016): *Biogeosciences*, 13(1):301–312.
- [3] Holtz LM, Wolf-Gladrow D and Thoms S (2014): *Journal of theoretical biology*, 364:305–315.
- [4] Holtz LM, Wolf-Gladrow D and Thoms S (2015): *Journal of Theoretical Biology*, 372:192–204.
- [5] Hopkinson BM, Dupont CL, Allen AE and Morel FMM (2011): *Proceedings of the National Academy of Sciences of the United States of America*, 108(10):3830–7.
- [6] Laws E, Popp B, Bidigare JRR, Riebesell U, Burkhardt S and Wakeham S (2001): *Geochemistry, Geophysics, Geosystems*, 2(2000).
- [7] Mackinder L, Wheeler G, Schroeder D, Riebesell U and Brownlee C (2010): *Geomicrobiology Journal*, 27(6-7):585–595.
- [8] Richier S, Fiorini S, Kerros ME, von Dassow P and Gattuso JP (2011): *Marine biology*, 158(3):551–560.
- [9] Rickaby REM, Henderiks J and Young JN (2010): *Climate of the Past*, 6(6):771–785.
- [10] Riebesell U, Revill AT, Holdsworth DG and Volkman JK (2000): *Geochimica et Cosmochimica Acta*, 64(24):4179–4192.
- [11] Romero MF, Chen AP, Parker MD and Boron WF (2013): *Molecular aspects of medicine*, 34(2-3):159–82.
- [12] von Dassow P, Ogata H, Probert I, Wincker P, Da Silva C, Audic S, Claverie JM and de Vargas C (2009): *Genome Biology*, 10(10):R114–R114.

Response to reviewer comments

McClelland et al.

We would firstly like to thank all three reviewers for their careful and insightful comments, and for their enthusiasm in our work. We are pleased to have the opportunity to address their concerns and improve this manuscript. There were some common points among these reviews which we have grouped and responded to together. The more specific points raised by the reviewers are individually addressed throughout the following document. Our responses are in blue sans-serif font below each comment.

Main points

1. Data in Figure 2

- R.#1** Figure 2. The legend was initially not clear to me. Perhaps it would be better to say "model output with data superimposed". I do have an issue with the statement about *E. huxleyi* $\delta^{13}\text{C}_{\text{cal}}$ increasing towards low DIC before sharply decreasing at very low values of DIC. This sharp decline is not clear from the figure. It would be better to expand the DIC scale at these low values to make this clear. Moreover, there is not any supporting data for these low DIC levels and it has been shown by a number of studies that calcification is inhibited at DIC values generally lower than 0.5 mM. How then, can the model predict any values for calcite $\delta^{13}\text{C}$ at low DIC?
- R.#3** The model appears to be biologically solid, but some of the model predictions are not apparent for the measured $\delta^{13}\text{C}$ data. For instance the high $\delta^{13}\text{C}$ at low DIC (Fig. 2) predicted from the model is not reproduced by the analysed coccolith of *E. huxleyi*. Similarly, the increase in $\delta^{13}\text{C}$ towards low DIC values is also hardly seen in the experimental Corg data. Therefore, the model data is clearly within uncertainty consistent with the experimental data, but the experimental data seem to provide no independent evidence for some features of the $\delta^{13}\text{C}$ vs DIC curves. Consequently, the model output (Fig.3) is somewhat questionable.

To address the concerns of the reviewers, we have now compiled and added all comparable data from the literature to Figure 2 [11, 2], and we have completely reworded the figure

caption to address the comments of Reviewer 1, and included their suggestions. We have also changed the X-axis to CO₂ concentration rather than DIC for consistency with the literature. As can be seen in our new Figure 2, the model successfully predicts the trends seen in these newly incorporated data. At high CO₂, calcite carbon isotopic compositions converge to abiogenic equilibrium calcite compositions (horizontal dotted line). At low CO₂ these compositions diverge (towards the left of the plot), trending towards heavy values in the low PIC:POC coccolithophore, *Pleurochrysis placolithoides*, and towards lighter values in the high PIC:POC coccolithophore *Calcidiscus leptoporus*. The slight increase in carbon isotopic compositions of *Emiliana huxleyi* calcite with decreasing CO₂ is indeed subtle ($\sim +1\%$), but it is present in our data set and also in that of Hermoso et al. (2016) [2] (also plotted).

Figure 2 shows the theoretically possible model output (i.e. where negative concentrations are not required) over CO₂ values ranging from 0 to 80 μ M. Reviewer 1 is correct that some of the included species have not yet been cultured at very low CO₂, or have been shown to be non-viable under such conditions. Thus, model predictions at very low CO₂ may not be verifiable in practice. However, we feel the limited experimental results do not warrant omission of the low CO₂ space as non-viable. We have therefore chosen to show model results for the entire CO₂ range, while now noting in the caption that parts of this model space are currently hypothetical in nature.

The rapid return to light values for the low PIC:POC coccolithophore species at extreme low CO₂ mentioned by Reviewer 1 is a description of the mathematics and is not theoretically realizable in most organisms. However, we feel it is important to make this point, because the curves are all described by the same set of equations, and critically, there is not a fundamental discontinuation between the trends towards heavy and light vital effects. The opposing behavior of the heavy and light groups is simply due to the relative magnitude of competing processes. Calcite carbon isotopic compositions are predicted to return to light values for coccolith calcite across all species - it just happens that this part of the curve exists in a realizable parameter space in the high PIC:POC species, but not in the low PIC:POC species. The figure caption has been reworded to reflect this.

For the organic carbon isotopic data, we have included the results from dilute batch culture experiments on *E. huxleyi* by Riebesell et al. (2000) [11], which show a strong increase in carbon isotopic compositions with decreasing CO₂, at values of CO₂ lower than our dataset. These data overlay our data very well, and clearly show the predicted increase in isotopic composition of *E. huxleyi* calcite at very low CO₂.

We have additionally provided a supplemental figure that includes C_{org} data from diatoms for reference as an end-member scenario for a closely related organism where PIC:POC = 0. We included all of the data from the literature we could find for dilute batch culture experiments on diatoms. The consistency of these data with our model is a testament to the robustness of the model. Their inclusion is appropriate because these are the data upon which the original classic description of the trends in organic carbon isotopic compositions are based, and onto which our model collapses at intracellular equilibrium and PIC:POC = 0. This supplemental figure also includes data from Rickaby et al. (2010) [10] whose calcite carbon isotopic compositions in

G. oceanica and in *C. pelagicus* perfectly overlay the others presented here. The organic carbon isotopic compositions in this study are questionable however, particularly those of *G. oceanica*, which are inconsistent with all other datasets in the literature, which unanimously describe an increasing magnitude of isotopic fractionation of carbon into organic matter with increasing CO₂. We therefore have included these data in the supplement for completion, but do not present them in the figure in the main manuscript. We have explained the omission of these data in the figure caption of Fig. S5.

The additional datasets taken from literature were not used for model calibration; indeed, they cannot be used for this purpose as they lack information on some of the environmental and physiological parameters needed to drive the model. The fact that these additional datasets nevertheless overlay model curves in Fig 2/S5, both within and beyond the range of data used for calibration, demonstrates the robustness of the model.

2. Bicarbonate transport

- R.#1** I would question the assumption (lines 407-415) that HCO₃⁻ movement across membranes is down-gradient and that Cl⁻ or Na⁺ concentration gradients are negligible. In seawater the concentration gradients for both of these ions across the plasma membrane are substantial. While the Cl⁻ electrochemical potential difference may be lower due to the negative membrane potential, there will be a very large inward electrochemical potential gradient for Na⁺. This gradient could well drive active HCO₃⁻ uptake and accumulation against a concentration gradient. Indeed there is evidence for active HCO₃⁻ accumulation operating in carbon concentrating mechanisms. How would be model behave if HCO₃⁻ uptake was driven uphill in this way?
- R.#2** The model assumptions are in general reasonable, however several in-depth clarifications are needed. In lines 412-415 it is stated: transmembrane concentration gradient for Cl⁻ and Na⁺ is negligible. This is highly unlikely as shown for example in the work by Taylor, 2009 on diatoms, which demonstrated of voltage-activated Na⁺ channels and the capacity to generate fast Na⁺-based action potentials in a unicellular photosynthetic organism and in *Coccolithus pelagicus* on the Cl-channel by Taylor and Brownley, 2003 proposing that 'major ion conductances play an essential role in membrane voltage regulation that relates to the unique transport physiology of these calcifying phytoplankton'. In addition, the concentration gradient of ions does not determine their trans-membrane flux, but the electro- chemical gradient does, i.e. one has to know the membrane voltage too.

We have changed the way that bicarbonate fluxes are formulated in this model, to address these points.

Reviewer 2's point that membrane voltage (or more specifically reversal potential) determines the direction and rate of the trans-membrane flux, and equilibrium concentrations, of charged ionic species in cells relative to their environment is true when the chemical species

in question traverses the membrane in its charged form. However, in coccolithophores, the current consensus in the literature is that the SLC4 family of bicarbonate transporter proteins is responsible for bicarbonate uptake [9, 7, 13]. In bicarbonate transport facilitated by the types of proteins that are probably present in coccolithophores, the HCO_3^- ion moves across a membrane paired to another ion with a favorable trans-membrane gradient. This can be a Na^+ ion moving in the same direction (symport), or a Cl^- ion moving in the opposite direction (anti-port), such that there is no net movement of charge [12]. As a result, the force driving the combined transport of HCO_3^- and its paired ion is dependent only on the trans-membrane gradients in both ions; it is independent of the cell membrane potential. In the revised model, this is described with a formulation based on the law of mass action.

The flux of bicarbonate symported across a membrane by electroneutral transporters is therefore proportional to the product of the concentrations of the bicarbonate ion and the Na^+ ion on the proximal side of the membrane. Similarly, the flux of bicarbonate anti-ported across a membrane is proportional to the product of the concentrations of the bicarbonate ion on the proximal side of the membrane and the Cl^- ion on the distal side of the membrane. As both Na^+ and Cl^- are more abundant outside the cell than in the cytosol, Cl^- -antiporting would inhibit the influx of HCO_3^- rather than enhance it. Therefore, we assume Na^+ -symporting to be the dominant means of bicarbonate entering the cell, as suggested by Reviewer 1. We factor this into the model very simply by introducing another universal constant (G), which represents the ratio of concentrations of Na^+ outside the cell to inside the cell; this is incorporated in the model as different effective permeabilities of the cellular membrane in each direction to bicarbonate. This new constant affects only the flux of bicarbonate from the external environment to the cytosol. This is, of course, a simplification, but we feel it is the most sophisticated and realistic description of bicarbonate transport that is reasonable, when trying to extract meaning from a model. We also effectively assume that gradients in Na^+ and Cl^- between intracellular compartments are small. We explain this new formulation throughout the main manuscript and in the supplement wherever bicarbonate transport is discussed. This new formulation allows for (but does not prescribe) net up-gradient bicarbonate transport into the cell.

3. CA in the external boundary layer

- R.#1** I am intrigued by the statement (line 469-471) that if future work concludes that external CA is absent in one or more species, it may be necessary to allow chemical and isotopic disequilibrium in the boundary layer. This begs the question of how the model will behave if the speciation of carbon in the boundary layer is significantly limited in this way. At least some speculation or model simulation would be useful in this context.
- R.#2** The assumption that 'CA is present in high enough concentrations that the external surface of the cell is in chemical and isotopic equilibrium with the bulk seawater' (lines 461-470) seems decided ad-hoc to justify the model since there is no evidence

that this is the case. It is important to elaborate on the other assumption on a model where CA is absent in one or more species and allows chemical and isotopic disequilibrium in the cellular boundary layer.

We would like to direct the reviewers to our discussion of this phenomenon in the supplementary material (starting from line 1472). Figure S3 explores the scenario that coccolithophores do not produce extracellular CA, where the boundary layer may be altered due to the sluggish reacto-diffusive kinetics that replenish the CO_2 pool. The concentration of HCO_3^- in the boundary layer will not change as it is in great abundance, but the boundary layer can become depleted in CO_2 . In this exploration, we have assumed that there is no isotopic fractionation by membranes, so the depletion of CO_2 does not cause a Rayleigh-type fractionation within this particular pool. When external CA is not present, and when growth rates are high, cells are large, and CO_2 concentrations are low, the boundary layer becomes significantly depleted in CO_2 . Trans-membrane fluxes of carbon species are determined by the concentration at the membrane, and thus the cell "sees" only this depleted environment, which is reflected in the isotopic compositions. As shown in Figure S3.F, the greatest theoretical depletion with no external CA could lead to ambient CO_2 being underestimated by as much as 50%.

The presence of external CA is not an ad-hoc assumption - In main manuscript (lines 496-536) and the supplementary section (lines 1574-1581), we cite a number of studies that have investigated extracellular CA in coccolithophores and concluded that the presence of external CA is likely. In all other modeling work to date [6, 1, 3, 4], equilibrium of the cell's surface with the ambient water is implicitly assumed - we make the same assumption, but state that the presence of external CA is the only reasonable mechanism whereby equilibrium with the ambient water may be maintained.

4. Proposed proxy approach

- R.#2** I found the last part of the paper on coccolith-based geochemical paleoclimate proxy for pCO_2 atm very speculative. It is known that the $\delta^{13}\text{C}$ in coccolithophores-producing alkenones is very limiting but in the paper I didn't find any robust suggestion for improving it. A robust proxy for coccolithophore PIC/POC for paleoreconstruction is not available at this moment.
- R.#3** While the significance of coccolithophores for the global C cycle is undoubted, their potential as geochemical proxy archive and the application of the presented approach is restricted by problems related to the isolation of pure mono-specific fractions of coccolith calcite from sediments and compound specific C isotope analyses of molecules that origin exclusively from a single coccolithophore species. To demonstrate the general applicability of the approach and increase the significance of the study, it would be desirable to provide $\delta^{13}\text{C}$ data of different coccolith and compound specific $\delta^{13}\text{C}$ org of coretop sediments, to illustrate that at least in modern sediments the methods yields consistent data.

The purpose of this last section of the paper is to present a new theoretical concept for constraining past CO₂ concentrations, and explain the current limitations. This is a direct implication of the increased understanding we have gained - other researchers will likely use our results thus, and so we feel this extension of the discussion is necessary. However, we have reworded the abstract and final paragraph to steer the reader away from the expectation that our primary aim is to calibrate and use a new proxy. Using isotopic analyses of coccolithophore biomass to estimate absolute pCO₂ in the past is currently associated with far higher uncertainty than has been portrayed in the literature - but our work constitutes a significant improvement in the theoretical underpinning of such an approach, and shows where improvements need to be made in order to achieve this goal.

We disagree with Reviewer #2 that the proxy approach we propose does not constitute a significant theoretical improvement in the alkenone CO₂ proxy. The alkenone CO₂ proxy as used in numerous high profile paleoclimate publications, relies on an assumption that the sole carbon source for coccolithophore cells is CO₂, acquired by passive diffusion. It also assumes that a coccolithophore cell is a single compartment, and it completely ignores calcite - the other preserved record of carbon isotopic compositions. At present, the alkenone CO₂ proxy requires the isotopic composition of contemporaneous seawater to be inferred from foraminifera or fish teeth. By contrast, the model presented here formally constrains the influence of CO₂ on the difference between the isotopic compositions of organic matter in large and small coccolithophores, calcite in large and small coccolithophores, and the difference between organic matter and calcite in a given species. This would potentially allow an external reference to be circumvented. Our model is of greater sophistication than the empirical relationship currently employed in the alkenone proxy, incorporating multiple intracellular compartments as well as HCO₃⁻ usage.

Regarding the suggestion by Reviewer #3 to provide a core-top investigation, we feel that the analysis of core-top samples would actually not be a useful addition to this study: Firstly, any calibration of the proposed proxy approach would ideally require a good constraint on the following parameters: cell size (r), growth rate (μ), PIC:POC, carbonate chemistry (e.g pH and CO₂) $\delta^{13}C_{org}$, $\delta^{13}C_{calcite}$ and $\delta^{13}C_{DIC}$. Without all of these parameters, the model is not fully constrained. Some of these parameters could be indirectly constrained; for example, the Strontium/Calcium ratio of coccolith calcite has been used as a proxy for growth rate, and the aspect ratio of coccoliths has recently been shown to be a potential proxy for PIC/POC. Indeed, use of such proxies is how one could use our mechanistic insight in a future palaeoclimate study. However, these proxies are currently immature, and have large associated uncertainties. In combination, the multiplied associated uncertainties would detract from our robust mechanistic findings. For coretops, $\delta^{13}C_{DIC}$ (an environmental parameter which would be required to infer carbon isotope vital effects) and the carbonate chemistry cannot be meaningfully constrained, as they have been in our lab cultures. This is due to the rapid rate of anthropogenic CO₂ and $\delta^{13}C_{CO_2}$ change over the past century, and the uncertainty in the time period represented by core tops (these samples suffer from loss of uppermost sediments during coring, and integrate multiple depths, due to varying degrees of burrowing: core top

coccoliths are typically dated to a few kyrs). Secondly, a meaningful calibration requires significant variability in the objective parameter. The variation in surface water CO₂ concentrations that the coccolithophores experience is due only to the effect of temperature on the Henry's law dissolution constant. The annual average CO₂ concentration variability in coccolithophore habitats worldwide is relatively small; much smaller than geological variation of pCO₂. It is for exactly these reasons that a core-top calibration of the widely used alkenone CO₂ proxy does not exist for example. There certainly will be huge value in applying this proposed approach for future down-core investigations of the past carbon cycle but first, better proxies are needed and this is what we wanted to highlight with our final perspective section. A core top study would add no value here, only uncertainty.

With respect to a possible proxy for PIC:POC mentioned by Reviewer 2, we have just had a manuscript published in *Scientific Reports* describing a new approach for reconstructing PIC:POC in these organisms from their ancient remains [8]. Further to this, obtaining compound-specific C_{org} isotopic compositions from monospecific fractions is absolutely possible, and we have done it. We are excited that Reviewer #3 pointed to these particular components of the problem. We have just had a paper accepted in *Nature Communications* describing the protocol for the extraction of polysaccharides from within the calcite lattice ("The uronic acid content of coccolith-associated polysaccharides provides an insight into coccolithogenesis and past climate", manuscript no. NCOMMS-15-16583). [Redacted]

Reviewer #1

This is a very interesting, detailed, clearly written and useful manuscript that presents a model based on generally well-supported assumptions for inorganic carbon transport and utilization in coccolithophores to understand the mechanistic basis of carbon isotope fractionation in coccolith calcite, including the distinction between isotopically "heavy" and "light" species. The model is constrained by culture experiments with three species of coccolithophore representing fast and slow growers. Subject to consideration of the comments below, this should be an extremely valuable contribution to understanding the interpretation of palaeoproxies and should significantly increase the understanding of the mechanistic interplay between photosynthetic and calcification carbon fluxes. It is likely to be of interest to a wide readership. I do have a couple of questions for clarification. My comments mainly relate to the basic biological assumptions rather than the construction of the model itself.

1. (Copied to Main Point 1.)
2. (Copied to Main Point 2.)
3. (Copied to Main Point 3.)
4. **Minor point:** The References are not presented in a consistent format. For example some have all authors listed, some only the first three and some have et al.

Response: Corrected. We have changed the format of references to follow Nature's guidelines for authors, specifically that: "All authors should be included in reference lists unless there are more than five, in which case only the first author should be given, followed by 'et al..'"

Reviewer #2

Isotopic fractionation in biologically precipitated calcite or 'vital effects' is a very fascinating and original topic enhancing the understanding of biomineralization, physiology and biogeosciences. In marine calcifying organisms present as fossils in the sedimentary record, this understanding can improve the application of geochemical isotopic proxies for paleoceanographic and paleoclimate reconstructions. Although the awareness about vital effects has been available for a few decades, there is still little work done on this topic. Coccolithophores as calcifying marine phytoplankton are a particularly interesting group

of calcifiers to study since they have a very long and continuous fossil record, are major carbonate producers, and some species like the cosmopolitan and abundant *Emiliana huxleyi* are largely studied in laboratory for physiological and chemical studies. In particular, they are known for their large oxygen and carbon vital effects in their coccolith calcite making them critical for understating biological fractionation. The paper by McClelland et al. is focusing on the carbon isotope vital effects in coccolith calcite and proposes a mechanistic understanding of the large species-specific isotopic fractionation. The key results relate to the fact that carbon (and oxygen) isotopic fractionation is largest when cellular carbon utilization is high, PIC/POC is low, and disappears when PIC/POC is high and carbon utilization is low. When the cellular carbon requirement is an appreciable fraction of the carbon pool to the cell, vital effects are largest. The main basic idea is that the isotopic composition of coccolith calcite is a function of the carbon fixation by photosynthesis and the calcification process having opposite effects on the isotopic composition of the coccolith vesicle bicarbonate pool. The effect of PIC/POC changes on the carbon isotopic composition of the organic matter is weak because the effect of calcification on the CO₂ isotopic composition in the chloroplast is far outweighed by that of carbon fixation. The proposed model is based on a competing Rayleigh-type fractionation process describing the partitioning of isotopes between two reservoirs. In the paper this is used within the cell affecting the bicarbonate pool in the intracellular coccolith vesicle where calcification of coccoliths is occurring. The kinetic isotopic fractionations of biologically-mediated processes can vary in magnitude, depending on several factors and in the case of coccolithophores includes species-specific metabolic transformations. The authors need to elaborate on the following statements:

- **lines 113-134:** HCO₃ as well as CO₂ usage. This is partially correct since it doesn't seem to be clear that 100% CO₂ can never work.

Response: We agree that it is not known that 100% CO₂ can never work - this is why we allow the model to constrain parameters that quantify the bicarbonate uptake required to fit the data - with the option of zero bicarbonate uptake explicitly allowed (See section S1). We have additionally reemphasized throughout the manuscript that the model can collapse back onto purely CO₂ supply by passive diffusion, if warranted by observations used to constrain the universal constants.

- **lines 134-136:** Published values for H⁺ concentration in different compartment. These values are published but are controversial since it's technically impossible to measure. The cytosolic pH is probably correct.

Response: We understand that these values are controversial, however, the values of Anning et al. are the only widely cited empirical (i.e. model-independent) values available in the literature for the coccolith vesicle and the chloroplast in multiple species. They are based on an established protocol, whereby cells were loaded with a pH-dependent fluorescent

dye prior to analysis with a confocal laser scanning microscope. The model is able to reproduce the same curves if other hypothetical pH combinations are prescribed, however, we are reluctant to present results that are based on non-empirical input.

- **lines 150-155:** 'Carbonate system reactions are quantifiable functions of intracompartamental DIC-species-specific concentrations, H⁺ concentration (pH), and the concentration of carbonic anhydrase (CA), the metalloenzyme responsible for catalysing the otherwise sluggish interconversion of CO₂ and HCO₃.' This is not clear. Clarify about the quantification. Where do all these come from? Ref? Ad- hoc assumption?

Response: The intention of this sentence is merely to list the interacting components of the equations that describe carbonate chemistry in any given pool. We have now referenced the widely used book by Zeebe and Wolf-Gladrow, and two more recent papers that discuss CA in depth. Other than pH and CA, all of these parameters are direct outputs of the model.

The value for intracellular CA was initially prescribed to be 0.1mM, which is close to that inferred by Hopkinson et al. (2011) for diatoms [5] - the model output shows that values of CA below this cause highly unrealistic model behavior no matter what parameters are used, and that concentrations above this have a fairly small effect on the model output. We therefore infer that the carbonate chemistry in the cell is close to equilibrium with respect to carbon. It cannot be in equilibrium with Oxygen (which takes 10x longer) because oxygen isotopic vital effects would not exist, and would instead simply reflect that of carbonate chemistry in equilibrium with water. There is therefore a fairly narrow range of CA (~ 0.1mM to ~ 1mM) that makes sense. We have amended the supplementary discussion to make these conclusions about CA clearer (line 2740-).

- (Copied to Main Point 2.)
- (Copied to Main Point 3.)
- It is interesting the comparison with oxygen isotopic fractionation in coccolith calcite when compared with carbon but the discussion provided is not very convincing (lines 658-676) and why the a plot showing the d13C/d18O is not provided? A better elaboration of the interpretation of oxygen versus carbon isotopic fractionation should be provided and compared also with the model presented for *Calcidiscus leptoporus* oxygen fractionation in Ziveri et al., 2012.

In *Calcidiscus leptoporus* and several planktic foraminifera species the calcite 18O is dependent on the seawater carbonate chemistry in addition to its dependence on temperature. The conceptual model based on the contribution of 18O-enriched HCO₃ to the CO₃²⁻ pool in the calcifying vesicle, could explain the [CO₃²⁻] effect on 18O for the different unicellular calcifiers (Ziveri et al., 2012). Please elaborate on this since the model do not fit with the proposed in this work. Also please consider that

there are evidence that calcification in *C. leptoporus* may occur in a highly alkaline environment (Gussone et al., 2007; Ziveri et al., 2012; Hermoso et al., 2014).

Response: It was an oversight on our part not to include a plot of oxygen isotopic compositions of calcite when preparing the manuscript. These data are now included (Fig S6), along with some aspects of the model output that informed our discussion of oxygen isotopes.

Whilst we agree that the oxygen isotopic composition of calcite is also a function of carbonate chemistry in addition to temperature, and indeed explain this in the discussion (lines 746-776), the purpose of this paper is exclusively to describe carbon isotopes. The mechanistic understanding behind carbon isotopic vital effects that the model provides is highly informative, but because the kinetic isotopic fractionation factors (KIFs) are not known for oxygen, for the hydration and dehydration of CO_2 and HCO_3^- respectively, oxygen isotopes cannot be modeled explicitly like carbon can. Any arguments for the origin of oxygen isotopic vital effects are therefore empirical and 'arm wavy' at best. We do not directly dispute the model of Ziveri et al. 2012, but also find their conclusions to be circumstantial. In particular, they propose the oxygen isotopic composition of the carbonate pool in the coccolith vesicle to be set by the ratio of HCO_3^- to CO_3^{2-} entering the coccolith vesicle. We do not agree that this ratio sets the isotopic composition of oxygen in the carbonate system of the coccolith vesicle because CO_2 forms such a large component of the transmembrane flux of DIC. The best we can do is to discuss whether oxygen isotopes are consistent with the model inferred fluxes that explain the carbon isotopic data. We feel that a cross-plot of oxygen vs. carbon isotopic vital effects would be of limited value.

We have now included an additional supplementary figure (S6) that shows the aspects of the model output that informed our qualitative discussion of oxygen. The pattern in oxygen isotopes are qualitatively the same as for carbon: as can be seen in this figure, *E. huxleyi* precipitates the heaviest calcite and *C. pelagicus* the lightest; and with the vital effects being largest at low CO_2 . However, because HCO_3^- is heavier than CO_2 in carbon but is lighter in oxygen, these trends in vital effects absolutely cannot be simultaneously explained by a change in the ratio of bicarbonate to CO_2 entering the cell or the coccolith vesicle. Similarly, one of the major sources of carbon isotopic vital effects that we discovered - the Rayleigh-type distillation within the chloroplast which causes a leakage of heavy carbon, is only relevant for carbon and is independent of oxygen. The up-regulation of HCO_3^- uptake at high cellular utilization results in larger cells taking up more HCO_3^- at a given CO_2 concentration, consistent with the smaller forms precipitating heavier calcite than the larger forms. We hypothesize that the universal drift towards equilibrium values at high CO_2 in oxygen is due to a faster rate of hydration and dehydration (for each hydration/dehydration cycle, the oxygen in the carbonate system becomes iteratively replaced by oxygen in water) - as shown in Fig S5.C.

- Since this paper aims to show the evolution of the carbon isotope vital effects in coccolith calcite, why were experiments on only *E. huxleyi* and *Gephyrocapsa oceanica* performed? Why not species with high PIC/POC (e.g. *C. leptoporus*, *H. carteri*)? And why 'evolution' when is proposing a mechanistic understanding?

Response: The word "origin" in the title does not refer to an evolutionary origin, but to a mechanistic origin. We like the title as it is, but if preferred by the editor, could change this to "source". Regarding other species with high and low PIC:POC please see our response to Main Point 1.

- (Copied to main point 4.)
- Finally, I found the paper interesting and possibly of interest for a publication in Nature. However, some major improvement needs to be done on the model assumptions, carbon versus oxygen vital effects and paleoclimate proxy discussion.

Reference:

Hermoso, 2014, *Cryptogamie Algologie* 35(4):323-352.
 Taylor A, 2009, *PlosONE* (<http://dx.doi.org/10.1371/journal.pone.0004966>),
 Taylor A and Brownley C, 2003, *Plant Physiol.* 2003 Mar; 131(3): 1391-1400,
 Ziveri P et al. (2012) *Biogeosciences*, 9, 1-8, 2012.

Reviewer #3

The manuscript "The origin of carbon isotope vital effects in coccolith calcite" by HLO McClelland et al. presents an interesting approach to describe and quantify cellular C budgets in coccolithophores.

The authors developed a detailed model, which is a great step towards a better understanding of the carbon budget of coccolithophores, important unicellular marine algae, which significantly contribute to the marine primary production. Nevertheless, there are two main issues related to the presented model and proposed proxy application.

Main points of criticism:

- (Copied to main point 1.)
- (Copied to main point 4.)

Detailed comments:

- The statement in lines 28/29: "..., which is consistent with oxygen isotopes" is misleading, as later in the text (line 650) it is stated that "...this model cannot currently be fully resolved with respect to oxygen isotopes"

Response: The compatibility of our model with trends seen in oxygen isotopes, as discussed in the text is important because contrasting theories - such as vital effects being driven purely by bicarbonate fluxes - are directly contradicted by these trends. We have changed this wording to " ..., which is compatible with trends in oxygen isotopes", which we hope makes our meaning clearer. Due to the large interest in oxygen isotopes in the paleoclimate community, we feel it is important to reference oxygen, as the conclusions presented here have highly relevant implications - through cellular fluxes of carbon species - even though a formal modeling of oxygen isotopes is not possible.

- Figure 1: subscripts are quite small, please consider to increase font size.

Response: We have increased the font size of the subscripts by two sizes.

- Lines 218 ff and Figure 2: indicate in Figure 2 or its caption, which coccolithophores represent high or low PIC/POC,

Response: We have now included this in the figure legend.

- respectively Figure S2: for consistency "Mu" should read " μ "

Response: Changed.

- Line 2681: please specify which figure ("Fig. ??") you intended to refer to

Response: Corrected.

References

- [1] Bolton CT and Stoll HM (2013): *Nature*, 500(7464):558–62.
- [2] Hermoso M, Chan IZX, McClelland HLO, Heureux aMC and Rickaby REM (2016): *Biogeosciences*, 13(1):301–312.
- [3] Holtz LM, Wolf-Gladrow D and Thoms S (2014): *Journal of theoretical biology*, 364:305–315.
- [4] Holtz LM, Wolf-Gladrow D and Thoms S (2015): *Journal of Theoretical Biology*, 372:192–204.
- [5] Hopkinson BM, Dupont CL, Allen AE and Morel FMM (2011): *Proceedings of the National Academy of Sciences of the United States of America*, 108(10):3830–7.
- [6] Laws E, Popp B, Bidigare JRR, Riebesell U, Burkhardt S and Wakeham S (2001): *Geochemistry, Geophysics, Geosystems*, 2(2000).
- [7] Mackinder L, Wheeler G, Schroeder D, Riebesell U and Brownlee C (2010): *Geomicrobiology Journal*, 27(6-7):585–595.
- [8] McClelland HLO, Barbarin N, Beaufort L, Hermoso M, Ferretti P, Greaves M and Rickaby REM (2016): *Scientific Reports*, 6(August):34263.
- [9] Richier S, Fiorini S, Kerros ME, von Dassow P and Gattuso JP (2011): *Marine biology*, 158(3):551–560.
- [10] Rickaby REM, Henderiks J and Young JN (2010): *Climate of the Past*, 6(6):771–785.
- [11] Riebesell U, Revill AT, Holdsworth DG and Volkman JK (2000): *Geochimica et Cosmochimica Acta*, 64(24):4179–4192.
- [12] Romero MF, Chen AP, Parker MD and Boron WF (2013): *Molecular aspects of medicine*, 34(2-3):159–82.
- [13] von Dassow P, Ogata H, Probert I, Wincker P, Da Silva C, Audic S, Claverie JM and de Vargas C (2009): *Genome Biology*, 10(10):R114–R114.

REVIEWERS' COMMENTS:

Reviewer #1 (Remarks to the Author):

The authors have addressed my concerns in the revised manuscript and in their response to referees' comments and the manuscript is significantly more clear in several respects. I have one small issue that should be clarified in the text:

The authors have correctly argued that Cl-HCO₃ exchange is an electroneutral process. However, we do not know whether this is the case for Na-HCO₃ exchange since different Na-coupled SLC4 transporters may be either electroneutral (1:1 stoichiometries) or may have Na:HCO₃ stoichiometries of 1:2 or 1:3 which would render them electrogenic and influenced by both membrane potential and the individual ion concentrations. I suggest that the authors provide the caveat "assuming electroneutral Na:HCO₃ exchange" in lines 179-183 but should acknowledge that this is not certain since the transporters have not been physiologically characterised in coccolithophores.

Reviewer #2 (Remarks to the Author):

The revised manuscript by McClelland and co-authors addressed the major concerns that I raised in the first submission. However, there are still a few points to clarify:

- Although I agree with the authors that improvement of the alkenone CO₂ proxy is a potential contribution of this paper, the proposed approach is only conceptual if not really proven in a down-core record by using the preserved record of carbon isotopic compositions in the coccolith calcite from the same alkenone producers. Several large species, such as *C. pelagicus* and *C. leptoporus*, are not alkenone-synthesising coccolithophores and are the species that are relatively easy to be separated in the fossil record.

The presented model is constraining the influence of CO₂ on the difference between the isotopic compositions of organic matter in large and small coccolithophores, calcite in large and small coccolithophores, and the difference between organic matter and calcite in a given species.

The authors's argument against a core-top calibration is a bit controversial, It relates to the lack of a good constraint on the several needed parameters (and a good temporal control of the most recent core-top sediments). If these proxies are currently immature, and have large associated uncertainties how could this approach be used in down-core material?

- It would be good for comparison with other published studies to provide the cross-plot of oxygen vs. carbon isotope data.

also:

. 15-17 However, coccoliths remain underused in palaeo-reconstructions, due largely to a lack of understanding of what controls their isotopic composition.

I would change into:

. The geochemistry of coccoliths remains underused in palaeoreconstructions, due largely to a lack of understanding of what controls their isotopic composition and to the difficulty to obtain species-specific samples.

As mentioned in my first review, this work is very interesting covering an original topic and enhancing the understanding of isotopic fractionation in biological systems, biomineralization,

physiology and biogeosciences.

P. Ziveri

Response to reviewers' comments

Our response to the reviewers comments are given in blue below their comments.

Reviewer 1

The authors have correctly argued that Cl-HCO₃ exchange is an electroneutral process. However, we do not know whether this is the case for Na-HCO₃ exchange since different Na-coupled SLC4 transporters may be either electroneutral (1:1 stoichiometries) or may have Na:HCO₃ stoichiometries of 1:2 or 1:3 which would render them electrogenic and influenced by both membrane potential and the individual ion concentrations. I suggest that the authors provide the caveat "assuming electroneutral Na:HCO₃ exchange" in lines 179-183 but should acknowledge that this is not certain since the transporters have not been physiologically characterised in coccolithophores.

- To address the comments of reviewer 1, we have inserted the following sentence: "We note that the electroneutrality of this process, and thus its independence from the membrane potential, is an assumption, because some Na⁺-coupled SLC4 transporters are known to have Na⁺:HCO₃⁻ stoichiometries other than 1:1, and these transporters have not yet been physiologically characterized in coccolithophores."

Reviewer 2

Although I agree with the authors that improvement of the alkenone CO₂ proxy is a potential contribution of this paper, the proposed approach is only conceptual if not really proven in a down-core record by using the preserved record of carbon isotopic compositions in the coccolith calcite from the same alkenone producers. Several large species, such as *C. pelagicus* and *C. leptoporus*, are not alkenone-synthesising coccolithophores and are the species that are relatively easy to be separated in the fossil record.

The presented model is constraining the influence of CO₂ on the difference between the isotopic compositions of organic matter in large and small coccolithophores, calcite in large and small coccolithophores, and the difference between organic matter and calcite in a given species.

The authors's argument against a core-top calibration is a bit controversial, It relates to the lack of a good constraint on the several needed parameters (and a good temporal control of the most recent core-top sediments). If these proxies are currently immature, and have large associated uncertainties how could this approach be used in down-core material?

- We agree with the reviewer that these are all aspects of the approach that must be overcome before this application can be widely used for accurately reconstructing CO₂. However, as mentioned in our previous response, the proposal of a new CO₂ proxy is not the primary purpose of this work – but it is an important implication.

The problems associated with the alkenone CO₂ proxy, and those that need to be overcome by any phytoplankton carbon isotope based CO₂ proxy can broadly be divided into:

1. Understanding the relationships between isotopic compositions and CO₂ in a simple or complex model;
2. Obtaining material for isotopic analysis that is representative of the model values (i.e. being able to measure the isotopic compositions that the model produces);
3. Constraining accurately all other required parameters (i.e. the parameter “b” in the alkenone proxy, or growth rate, cell size and PIC:POC in our approach).

The present study deals with just the first of these aspects, and we have taken great care to emphasize the conceptual nature of our proposed proxy approach, both in the last sentence of the abstract, and throughout the last paragraph of the manuscript.

We have also now included the following sentences in the last paragraph of the discussion, which addresses the second of the above aspects:

“ With paired measurements of Ec and Eo from analysis of ancient sediments, in theory our model can be used to iteratively search parameter space to minimise the misfit between observed and predicted isotopic compositions, and thus simultaneously predict the most likely values of PIC:POC and Tau of these ancient organisms. We recently extracted acidic polysaccharides from within the calcite lattice of large ancient coccoliths (Lee et al. 2016, Nature communications), which opens up the possibility of isotopically characterizing non-alkenone producing species, and may, in the future, supersede alkenones as the target molecule for organic carbon isotopic analyses. “

Lastly, regarding the third aspect, the information we present here is essential for providing a framework to communicate to wider paleoclimate research community, the need for new approaches and proxies that are necessary before accurate CO₂ barometry (and accurate calibration) is possible using this technique.

- It would be good for comparison with other published studies to provide the cross-plot of oxygen vs. carbon isotope data.

- We do not agree that a cross-plot of carbon vs oxygen isotopes is helpful in this instance. We do not claim to yet understand how oxygen isotopes work in these organisms. The system is more complicated than suggested by the vaguely positive correlation between carbon and oxygen isotopic compositions across strains (see figure below). By presenting this figure, we may imply a causal relationship to the reader. At this reviewer’s request, we have provided this figure, which include oxygen isotopes as a

separate system plotted against CO₂ and shows those aspects of model output that we use to discuss possible mechanisms behind oxygen isotopic fractionations. All new data are available as a supplementary dataset for specialist researchers who wish to investigate these empirical relationships further.

also:

. 15-17 However, coccoliths remain underused in palaeo-reconstructions, due largely to a lack of understanding of what controls their isotopic composition.

I would change into:

. The geochemistry of coccoliths remains underused in palaeoreconstructions, due largely to a lack of understanding of what controls their isotopic composition and to the difficulty to obtain species-specific samples.

- We have now removed the second part of our original sentence to reduce the length of the abstract. This change makes the change the reviewer suggests here unnecessary. The sentence now reads: “However, unlike the shells of foraminifera, their zooplankton counterparts, coccoliths remain underused in palaeo-reconstructions.”